# NOISE-GUIDED UNSUPERVISED OUTLIER DETECTION

## ABSTRACT

Over the past decade, we have witnessed enormous research on unsupervised outlier detection techniques, ranging from statistical models to recent deep learning-based approaches. Existing approaches generally limit their discussions to unlabeled data mixed with normal (inlier) and abnormal (outlier) data, which constitute only a tiny fraction of the whole value space. Such approaches tend to fall into the local optimum of a specific subspace and hardly generalize to diverse datasets. This paper proposes a novel end-to-end Noise-guided unsupervised Outlier Detector (NOD), which infers the anomaly score of the entire value space via a simple MLP to learn the difference between samples and uniform noise. We further theoretically prove that the learned classifier can separate outliers from inliers with limited samples. Extensive experiments show that NOD significantly advances UOD performance in 22 diverse real-world datasets by an average of 30.6% ROC_AUC against 11 state-of-the-art counterparts without dataset-specific tuning. The merit is of paramount importance for UOD due to the lack of labeled data for supervision.

## 1 INTRODUCTION

Outliers are data points, events, or observations that deviate from the bulk of the data. Outliers are important as they often reflect variability, experimental error, or novelty. Outlier detection is essential in various applications, such as network intrusion detection (Weller-Fahy et al., 2014), malicious behavior detection(Yu et al., 2016), machine failure (Riazi et al., 2019), etc. As a sufficient amount of outliers and correct labels is often expensive to obtain, outlier detectors typically need to handle unlabeled data containing a mixture of inliers and outliers. Due to the lack of labels, designing unsupervised outlier detectors (UODs) is rather challenging (Pang et al., 2021; Wang et al., 2019).

From the perspective of real-world applications, inliers are generated from certain mechanisms of an underlying system with specific internal structures. It is vital to develop an accurate model to describe these structures to separate inliers from outliers. Due to the lack of labels, traditional UODs often develop predefined inlier/outlier models and an arbitrarily anomaly score function; e.g., inliers have more neighbors than outliers (Breunig et al., 2000), the clusters of inliers are larger and denser (Thang & Kim, 2011), outliers can be divided into a single subspace faster (Liu et al., 2008) or are farther away from a stable statistical model (Shyu et al., 2003; Almardeny et al., 2020), etc. As real-world datasets come with highly diverse modes of outliers and/or inliers, solutions with predefined profiles often suffer from volatile performance across different datasets.

Many DNN-based methods have recently been introduced to learn rather than "design" the internal structures (Pang et al., 2021) due to its powerful modeling capabilities. However, the most recent approaches, for example, EBGAN (Zenati et al., 2018) and SSD (Sehwag et al., 2020) are generally rely on the availability of pure inliers in the training data with certain self-supervised tasks. Their settings differ significantly with UOD and are denoted as self-supervised(Sehwag et al., 2020) or semi-supervised ODs(Wang et al., 2019)(both SSOD for short).

The absence of inlier and outlier labels presents great challenges in adopting DNNs to learn only the inlier manifold with mixed samples, which results in much less progress than SSOD (Wang et al., 2019). State-of-the-art methods are mainly limited to analyzing the different characteristics of inliers and outliers during representation learning. For example, deep AutoEncoder (AE)-based approaches (Kingma & Welling, 2013; Chen et al., 2017) assume that outliers are harder to reconstruct. However, this is not always true, especially in scenarios with a high proportion of outliers. Other

researchers design UOD with domain-specific knowledge, e.g. $E^3$ outlier (Wang et al., 2019) with image rotations. They can hardly be used in general applications.

Actually, one fundamental observation in UOD is that inliers are denser distributed than ouliers (Thang & Kim, 2011). Thus, the problem becomes how to estimate a probability density distribution(density distribution for brevity) function $a(\cdot)$ in an n-dimensional Euclidean space $\mathbb{R}^n$. Theoretically, it is possible to estimate the probability density $a(\cdot)$ by training a classifier to discriminate between samples and any artificially generated noise $b(\cdot)$(Sugiyama et al., 2012). This proof requires infinite samples for both samples and noises. Two recent works generate *synthetic* outliers(noises) close to samples via generative adversarial networks (GANs) in SO-GAAL and subspace perturbation in LU-NAR (Goodge et al., 2021), as Gutmann & Hyvärinen (2012) points out that $b(\cdot)$ should theoretically be close to $a(\cdot)$. However, in practice, generating synthetic outliers with a similar distribution with finite samples is very difficult and can often result in the model collapse in classification.

In reality, due to the limitations of the underlying physics, outliers and inliers generally lie in a relatively small number of manifolds with different density distributions throughout the value space $\mathbb{R}^{dim}$ (Lee, 2013). Due to the highly unbalanced nature of the outliers, the outliers should be generally sparser distributed than inliers and the anomaly score in $\mathbb{R}^{dim}$ should be smooth. Then, the problem becomes how to effectively estimate the distribution. From *Principal of Maximum Entropy* proposed by Jaynes (1957), noises with uniform prior probability density in $\mathbb{R}^{dim}$ have the maximum entropy, and thus essentially have little chance of resembling structured input(Goodfellow et al., 2016). From this point of view, a randomly given point in $\mathbb{R}^{dim}$ should have a very high probability of being an "outlier" than most samples in the unlabeled data. Compared to the synthetic outliers generated by GANs or subspaces, random noise can act as a trustworthy and stable reference for a sample's anomaly degree in the whole data space and demands few assumptions towards the inliers' distribution. Inspired by this observation, this paper proposes NOD, an end-to-end Noise-guided unsupervised Outlier Detector that learns the anomaly score with a simple binary classifier that is trained to distinguish between samples and uniform noise generated throughout the value space. The main characteristics of NOD are as follows:

1. **Simple and efficient.** NOD is an extremely simple end-to-end DNN-based unsupervised outlier detection solution that uses pure uniform noise, MLP, and an off-the-shelf backpropagation mechanism. This model can be directly used for outlier detection in the whole value space. NOD can take advantage of GPU acceleration to easily process datasets with high-dimensional features and a large number of samples.

2. **Theoretically sound.** We translate the UOD problem into a density estimation problem with a classification and relax the theoretical conditions from unlimited data to limited samples if the classifier is subject to the "smooth prior" restriction.

3. **Effective and robust.** Extensive experiments performed on 22 datasets show that NOD significantly outperforms 11 state-of-the-art detectors with significant edges and scores $1.1\% \sim 74.4\%$ higher on average ROC_AUC. Further parametric analysis shows that NOD is insensitive to its few hyperparameters.

## 2 RELATED WORK

**Classic outlier detection.** Classical outlier detection algorithms are often designed to estimate data distributions and design custom anomaly functions to determine sample deviation degrees. kNN (Ramaswamy et al., 2000) takes the distance between the samples and their k-th nearest neighbor as the anomaly score, and LOF (Breunig et al., 2000) calculates the average density ratio between the samples and their k-th nearest neighbor to assess the rarity of the sample. COPOD (Li et al., 2020) and ECOD (Li et al., 2022) hypothesize that inliers have a consistent distribution pattern and assess the anomaly score of the sample by calculating the tail probability from either a global or single-feature point of view. IForest (Liu et al., 2008) assumes that outliers would be divided into separate subspaces earlier than inliers during dividing hyperplanes. These approaches make respective designs/assumptions in the anomaly degree calculation which may not be valid across different datasets.

**Self-supervised outlier detection.** SSODs generally assume that the inliers are on a low-dimensional manifold and use different methods to learn this manifold. For the one-class classification stream,

One-Class SVM (Schölkopf et al., 2001) trains an SVM-based one-class classifier on inliers, and Deep SVDD Ruff et al. (2018) learns the output of the network into a hypersphere of minimum volume. DROCC (Goyal et al., 2020) tackles the representation model collapse problem by assuming that inliers lie on a well-sampled, locally linear low-dimensional manifold. Other streams, such as AnoGAN (Schlegl et al., 2017), use Generative Adversarial Networks (GAN) Goodfellow et al. (2014) to learn the inliers. Schlachter et al. (2019) propose contrastive learning in learning inliers. In particular, some SSODs, e.g. DROCC Goyal et al. (2020) and GOAD Bergman & Hoshen (2020), can be tailored to UOD by replacing pure inliers with mixed samples at the cost of a certain performance deterioration (see Appendix D).

**DNN-based unsupervised outlier detection.** Most UOD approaches use AE-based techniques (Hawkins et al., 2002; Xia et al., 2015) based on the assumption that inliers can be decoded better than outliers (Kingma & Welling, 2013). Later works use different ways to improve the robustness of AE, e.g. AE ensembles (Chen et al., 2017), robust principal component analysis (Zhou & Paffenroth, 2017), variational AE (Abati et al., 2019) and DAGMM (Zong et al., 2018). REPEN (Pang et al., 2018) focuses on learning representation to compress high-dimensional features into low-dimensional representations while still relying on off-the-shelf UOD methods. Recent approaches adopt *synthetic outliers*, e.g., Liu et al. (2020) use GAN and Goodge et al. (2021) use subspace perturbation to generate synthetic outliers. However, these synthetic outliers often overlap with positive samples, resulting in frequent model collapses. Wang et al. (2019) designs $E^3$Outlier with the CV-specific transformation tasks which does not apply to general tabular data. Recently, Qiu et al. (2021) proposed Neutral AD that uses contrastive learning to learn the invariants that exist among different learnable transformations. However, appropriate transformation functions must be used for different datasets for optimization, which is problematic in UOD.

## 3 PROPOSED APPROACH

As discussed above, the core of UOD can be seen as a density estimation problem, which is notoriously difficult in high-dimensional data(Liu et al., 2021). Sugiyama et al. (2012) points out that the density estimation problem, which is an unsupervised learning problem, can be solved by supervised training a classifier to distinguish between samples and any known distribution with unlimited samples.

### 3.1 NOTATIONS AND TRAINING OBJECTIVE

Given a dataset $X = [x_1, x_2, ..., x_n] \in \mathbb{R}^{n \times dim}$ with unobserved labels $Y = [y_1, ..., y_n]$, where $y_i$ is 0 or 1. $y_i = 0$ indicates that the sample is an inlier, and $y_i = 1$ indicates that it is an outlier. $n$ denotes the total number of samples and $dim$ is the feature dimension. Outlier detection aims to find an anomaly score function that maps $X$ to their unobserved labels without prior knowledge of $Y$. We denote $X_n, X_o$ as the set of inliers and outliers ($X = X_n \cup X_o$). $X^-$ is an auxiliary noise set generated from a uniform distribution in the value space $\mathbb{R}^{dim}$ of $X$. The core idea of NOD is to train a binary classifier $f(x)$ to separate inliers $X_n$ and outliers $X_o$ in the presence of auxiliary uniform noise. To achieve this, NOD leverages a positive sample set $x_i \in X$ with pseudo-labels of 0 and a negative sample set $x_k \in X^-$ with pseudo-labels of 1. For the binary classification problem, we use binary cross-entropy loss to optimize the classifier $f(x)$ by minimizing the loss $\mathcal{L}_f$:

$$\mathcal{L}_f^n = -\Big( \sum_{i=0}^{|X|} \log(1 - f(x_i)) + \sum_{k=0}^{|X^-|} \log f(x_k) \Big). \tag{1}$$

When we have arbitrarily large samples, the weak law of large numbers shows that the objective function $\mathcal{L}_f^n$ converges in probability to $\mathcal{L}_f$:

$$\mathcal{L}_f = -\big( E_X(\log(1 - f(x))) + E_{X^-}(\log f(x^-)) \big). \tag{2}$$

$f(x)$ is the output of the classifier with input $x$ and is the predicted anomaly score of the sample $x$. As demonstrated in (Gutmann & Hyvärinen, 2012), we can obtain the optimal classifier $f^*(x) \approx p(y = 1|x)$ after minimizing the loss $\mathcal{L}_f$ with logistic regression(detailed in Appendix A). However, in practice, we only have limited samples. The following section proofs that we can still have a classifier make density estimation with limited samples if appropriate conditions are placed.

## 3.2 THEORETICAL ANALYSIS

To distinguish inliers and outliers with limited samples, two restrictions are placed, one assumption on the datasets and another on the optimizer. We then provide a simplified proof of the correctness of NOD. A more comprehensive proof can be found in Appendix A.

**Assumption 1. [Distribution assumption]** *Outliers are sparser distributed than inliers and should be sufficiently distant from any inlier.*

Due to the highly unbalanced nature of the sample, we assume that outliers are sparser distributed than inliers and nonoverlapping with inliers. Without this assumption, it would be very hard to differentiate between inliers and outliers. This assumption has been adopted in density-based studies. However, the difficulty lies in how to effectively and efficiently estimate the density of high-dimensional data, due to the "curse of dimensionality". Many UOD calibrate the anomaly score based on the localized distance/density estimation to reduce computation cost. It is difficult for them to use samples beyond their scope. Fig. 1(a) of a toy sample shows the limitations of kNN in the localized calculation.

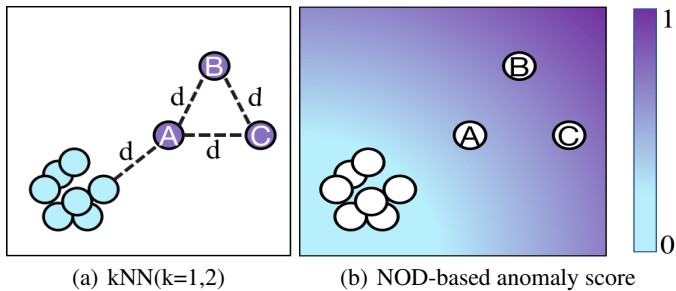

(a) kNN(k=1,2)    (b) NOD-based anomaly score

Figure 1: Anomaly score with kNN and NOD on the 2D dataset. Sample A is close to the normal cluster and with lower score than B,C. However, kNN(k=1,2) assign same socre to A,B and C.

Certainly, the sparse assumption may not be valid for outliers that are clustered together. However, we argue that we can only effectively address these small clusters with domain-specific knowledge. Section 4.3 contains a discussion on the effects of clustered outliers.

**Proposition 1. [Smoothness Prior]** Given a distance function $d$, $\forall x_i, x_j \in \mathbb{R}^{dim}$, if $d(x_i, x_j) < \varepsilon$, $\forall \varepsilon, |f(x_i) - f(x_j)| < M\varepsilon$. $f(\cdot)$ is the anomaly score of the sample $x$ with $0 \leq f(\cdot) \leq 1$, and $M$ is a positive constant subject to the Lipschitz condition.

To estimate accuracy with a limited number of samples, we impose a restriction on the optimizer so that the learned outlier score function is smooth, in accordance with underlying physical laws. This restriction specifies that the change rate of $f(x)$ across the entire value space is below a certain threshold and has been widely used in the design of many optimizers, especially those optimizers used in DNNs to estimate a smooth function, e.g. SGD (Bottou & Bousquet, 2007).

**Lemma 1.** When the value space is limited, using a limited amount of uniform noise, it is ensured that $\rho(x_i) > \rho(x_k) > \rho(x_j)$, where $x_i \in X_n, x_j \in X_o, x_k \in X^-$, and $\rho(\cdot)$ is the density function.

This lemma states that by constructing uniform noises, we can ensure that their densities fall between the maximum density of outliers and the minimum density of inliers. This can be achieved with a limited number of points, provided that the value space is limited. In practice, we can scale the value space into a unit hypercube $\mathbb{R}^{dim}$ with a MinMax scaler. With *Lemma* 1 and Assumptions 1, 2, we have that $\forall x_i \in X_n, x_j \in X_o, x_{j1}, x_{j2} \in X_o, \exists x_k, x_{k1}, x_{k2} \in X^-$, then, $d(x_i, x_j) > d(x_i, x_k), d(x_{j1}, x_{j2}) > d(x_{k1}, x_{k2})$.

**Lemma 2.** Let $D = \max_j \left( \min_k d(x_j, x_k) \right)$, where $x_j \in X_o, x_k \in X^-$. There exists an optimized $f^*(x)$ with respect to Equ. 1 that satisfy $\forall x_j \in X_o, f^*(x_j) \geq 1 - MD, MD < 1$.

This lemma shows that the learned optimized function always gives the outlier an anomaly score bigger than a certain positive value.

**Theorem 1.** Each predicted value of the outlier is higher than each predicted value of the inlier. $\forall x_i \in X_n, x_j \in X_o$, it holds that $f^*(x_j) > \lambda > f^*(x_i)$, where $\lambda$ is a boundary value.

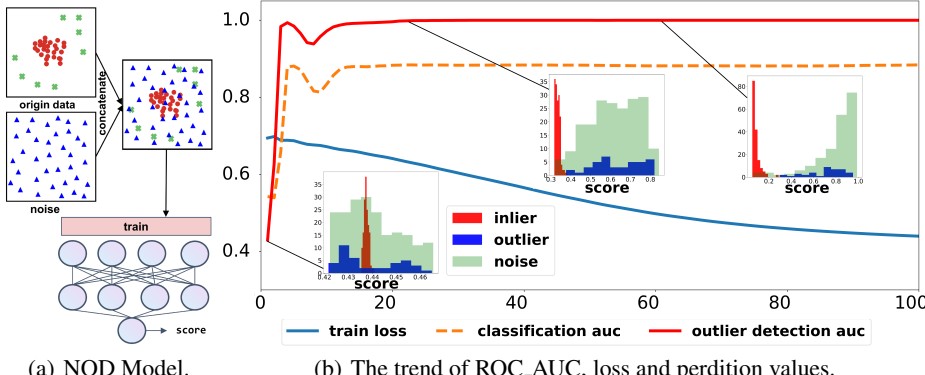

(a) NOD Model.  (b) The trend of ROC_AUC, loss and perdition values.

Figure 2: The NOD model and an exemplar training process. Inliers are red dots, outliers with green crosses, and noise with blue triangles. In (b), the histograms are the distribution of anomaly scores.

**Proof** Due to the high density of inliers, $\forall x_i \in X_n$, when $\rho(x_i) \to +\infty$, we have $f^*(x_i) \to 0$. Thus, there exists a density value $\rho_0$, s.t. $\forall x_i \in X_n$, we have $f^*(x_i) < \tau$. According to *Lemma 2*, it is possible to learn a classifier $f^*(\cdot)$ that satisfies the following conditions: $\forall x_i \in X_n, x_j \in X_o, f^*(x_i) < \tau \le \lambda \le 1 - MD \le f^*(x_j)$. For example, when $M < \frac{1}{2D}, \tau = 0.5$. Therefore, *Theorem 1* holds.

This theorem establishes that the anomaly scores of outliers are higher than those of inliers. If we have an outlier ratio, the classifier $f^*(x)$ can distinguish between $X_n$ and $X_o$. Fig. 1(b) shows the anomaly score distribution in the toy example. It clearly shows that NOD can effectively balance the impact of both local and remote samples with the support of uniform noise. The anomaly scores span the entire value space and exhibit a gradual increase as the points move farther from the inlier center. Thus, NOD can identify the degree of anomaly of sample B, and C is higher than that of sample A.

### 3.3 NOISE-GUIDED OUTLIER DETECTION

Based on the analysis in Sec. 3.2, the outlier detection problem is converted into a binary classification problem with the original data as positive samples (labeled 0) while the generated noise as negative samples (labeled 1). We scale the samples to [0, 1] with a MinMax scaler in each feature dimension. Then, we generate noises that uniformly cover the whole space of this unit hypercube.

Fig. 2(a) shows that a simple MLP can be trained to identify between positive and negative samples with any optimizer that satisfies the *smoothness prior*. The loss function is as Equ. 1. Here, we donot adopt the typically validation-based earlystop, as the sampling process might change the distribution of samples, especially on small datasets. Here, we design a new curve-based earlystop. The training process is stopped if when the change of training ROC_AUC is less than certain threshold for certain epochs(0.01 for 100 epochs in experiments). Here, the typically adopted validation-based earlystop is not used, as it might change the distribution of samples. Details are in Sec. 4.6. After training, the model outputs an anomaly score for any point in $\mathbb{R}^{dim}$.

NOD has two important characteristics to alleviate the density estimation problem in high-dimensional space: 1) We generate a set of noise uniformly distributed in the value space, which gives the classifier stable reference points across the value space. 2) The DNN's of-the-shelf smooth prior optimizers are the go-to method for large-scale optimization problems in data science. There are many relevant theoretical works; e.g., Arous et al. (2022) proved the capabilities of SGD in high-dimensional data. Fig. 2(b) depicts an example training process on a synthetic 2D data set (in supplementary data). In the initial stage (leftmost histogram), the anomaly scores between positive and negative samples are mixed. As training progresses, the distance between positive and negative samples gradually widens. The gap between inliers and outliers continues to widen during training. After about 60 epochs, the rightmost histogram displays a clear boundary between inliers and outliers.

### 3.4 NEGATIVE SAMPLE GENERATION

Several methods can be applied to generate uniform noise in a given space. 1) Poisson disk sampling ensures that all samples are at least $r$ distance apart for some user-supplied density parameter $r$. Avoid being too close to two samples. 2) Uniform Random (UR), uses a uniform probability distribution to generate samples with random values from 0 to 1 in the $dim$ dimensional space. The Fast Poisson Disk (FPD) implementation Bridson (2007) can be used to generate $n$ samples with $\mathcal{O}(n)$ complexity. Its running speed drops significantly as $dim$ increases. UR is much faster compared to FPD.

Our empirical analysis observes no significant performance gap between FRD and UR, although UR cannot guarantee uniformly spaced negative samples. Taking into account the run time in higher dimensions, the UR is adopted for negative sampling. The impact of different numbers of negative samples and other types of noise are in Sec. 4.6 and Appendix E.

## 4 EXPERIMENTS

### 4.1 EXPERIMENT SETTINGS

In this section, we evaluate the performance of NOD from different perspectives. The code, datasets, and detailed results for more than 20 baselines are in the supplementary material for reference.

**Datasets**. 22 real-world datasets from ODDS (Rayana, 2016) and DAMI (Campos et al., 2016), are used for evaluation. They cover a wide range of anomaly cases with diversity in dimension, data volume, and anomaly rate. The datasets are normalized to [0,1] by Min-Max Normalization. We generate the same size of noise as the positive sample $|X|$.

**Compared baselines.** Eleven strong baselines are compared, including *three classic baselines:* density-based LOF (Breunig et al., 2000), IForest (Liu et al., 2008) and ECOD (Li et al., 2022); eight *DNN-based baselines:* Deep SVDD (D_SVDD) (Ruff et al., 2018), DROCC (Goyal et al., 2020), ICL (Shenkar & Wolf, 2021), Neutral AD (N_AD) (Qiu et al., 2021), SO-GAAL (Liu et al., 2020), LUNAR (Goodge et al., 2021), REPEN (Pang et al., 2018) and DAGMM (Zong et al., 2018). For LOF, ECOD and IForest, we use the implementations from PyOD Zhao et al. (2019), a popular and open-source Python library for Outlier Detection with implementations of many well-known outlier detection methods. For others, we use the code and default settings given in their papers. For D_SVDD, DROCC, ICL, N_AD and LUNAR, they have both SSOD and UOD implementations. We use their UOD setting by using the original datasets containing both inliers and outliers for model training. In Appendix D, their performance under both SSOD and UOD settings is provided.

**Evaluation Metrics and Parameter Settings.** ROC_AUC (Hendrycks & Gimpel, 2016) and its standard deviation are used for the evaluation. Twenty trials are carried out, and those average values are taken as the final results. The results of the F1 score are in Appendix C. Average ranks are used to describe the overall performance of the outlier detectors across different datasets. All tested outlier detectors from PyOD use their default settings, which are optimized for most datasets. Others are the same as their original papers. More details of the settings can be found in Appendix C. NOD uses a two-layer MLP with a sigmoid activation function as the classifier. The dimensions of the hidden layer are consistent with the features. The SGD optimizer is adopted with a learning rate of 0.01 and L2 norm $10^{-6}$. NOD is trained with a maximum of 10,000 epochs with curve-based earlystop.

### 4.2 PERFORMANCE COMPARISON ON TABULAR DATA

Table 1 shows the experimental results on 22 datasets. NOD has very stable performance and achieves better ROC_AUC in almost all data sets with the best averaged ROC_AUC (83.0%), the best Rank_avg(2.5) and the best Rank_std(1.1). In comparison, results also show that outlier detectors designed with predefined assumptions normally suffer significant performance variance on different datasets, e.g., ECOD, LOF and IForest. Compared to other density-based methods that use both inliers and outliers to calculate the density, NOD uses negative samples as stable reference points and uses SGD to learn rather than assume the difference in the distribution. Appendix I shows the performance of other density-based detectors. This difference partially explains the significant performance advantage of NOD over LOF, which is also a density-based solution. However, most DNN-based UOD solutions underperform in these datasets, as it is difficult to learn internal structures

Table 1: Averaged ROC_AUC (%) and rank (in parenthesis) of 20 independent trials. The highest score is bolded and the second is underlined. Datasets are sorted in ascending order according to # features. F1 holds similar trends. Detailed results for 20 baselines are given in Appendix C.

| Dataset | LOF | ECOD | IForest | D_SVDD | DROCC | ICL | N_AD | SO-GAAL | LUNAR | REPEN | DAGMM | NOD |
|---|---|---|---|---|---|---|---|---|---|---|---|---|
| pima | 53.8 | 51.7 | **67.3**±0.9 | 48.8±10.8 | 48.2±30.2 | 51.5±18.2 | 49.9±1.5 | 50.8±1.2 | 50.5±0.1 | 64.4±2.8 | 59.0±5.0 | 62.9±3.0 |
| breastw | 38.3 | 99.1 | 98.7±0.2 | 78.0±19.6 | 46.8±31.0 | 82.7±3.7 | 70.4±2.0 | 97.6±0.3 | 49.4±0.1 | 98.8±0.3 | 96.8±2.7 | **99.3**±0.2 |
| WBC | 83.0 | 99.0 | 99.0±0.2 | 89.4±14.2 | 53.8±31.8 | 73.4±11.6 | 85.8±2.3 | 95.7±0.5 | 47.1±0.3 | **99.2**±0.2 | 84.3±13.0 | 98.8±0.2 |
| wine | 99.8 | 71.0 | 79.2±3.7 | 42.3±28.2 | 47.9±31.6 | 49.2±2.4 | 79.3±4.6 | 51.1±1.25 | 30.0±0.6 | **99.9**±0.1 | 95.5±9.2 | 97.2±1.2 |
| HeartDisease | 50.0 | 58.8 | 62.2±1.2 | 48.8±17.7 | 38.6±26.8 | **99.9**±0.1 | 46.2±2.4 | 42.4±8.4 | 47.9±0.3 | 66.0±2.7 | 77.1±4.8 | 67.1±3.8 |
| pendigits | 47.9 | 90.9 | 94.4±1.1 | 49.3±25.6 | 44.7±30.4 | 52.9±2.0 | 78.6±4.9 | 66.2±9.7 | 56.4±0.1 | **97.7**±0.3 | 91.7±3.6 | 91.6±1.8 |
| Lymphography | 97.6 | 99.5 | **99.9**±0.1 | 54.6±25.2 | 56.4±31.2 | 65.8±5.0 | 83.0±9.3 | 94.9±8.7 | 25.3±1.4 | 99.1±0.5 | / | 99.7±0.3 |
| Hepatitis | 62.6 | 78.6 | 69.4±1.9 | 50.3±18.9 | 39.7±25.8 | **90.6**±6.8 | 39.2±11.9 | 44.3±9.7 | 46.5±4.9 | 76.8±5.6 | 60.7±12.5 | 69.7±3.4 |
| Waveform | 73.4 | 72.0 | 70.8±1.8 | 54.4±17.4 | 53.8±35.6 | 53.8±6.5 | 76.1±2.1 | 33.8±3.0 | 49.5±0.1 | 78.0±4.7 | 60.8±12.1 | **80.7**±4.8 |
| wbc | 93.0 | 90.0 | 93.7±0.8 | 64.1±26.8 | 45.6±31.0 | 55.0±9.6 | 85.7±2.6 | 12.3±6.7 | 42.6±0.3 | 95.8±0.5 | 94.4±3.5 | **95.9**±0.8 |
| WDBC | 98.2 | 91.7 | 93.5±0.9 | 47.3±27.5 | 62.6±30.1 | 90.7±1.3 | 96.6±0.9 | 50.4±0.3 | 47.6±0.3 | **98.9**±0.3 | 89.0±13.3 | 97.4±0.5 |
| WPBC | 51.8 | 48.0 | 49.0±1.5 | 49.5±6.6 | 54.2±36.6 | **61.7**±5.7 | 43.9±3.3 | 50.2±4.0 | 47.8±0.2 | 52.4±2.2 | 55.6±5.8 | 57.8±1.3 |
| satimage-2 | 53.2 | 97.3 | 99.4±0.1 | 61.8±32.8 | 58.6±30.6 | 94.2±2.3 | 97.2±0.7 | 44.8±10.1 | 55.4±0.1 | **99.9**±0.1 | 88.7±10.3 | 99.5±0.1 |
| satellite | 54.0 | **74.6** | 70.8±1.7 | 53.6±13.0 | 50.4±33.6 | 56.9±3.9 | 70.2±2.2 | 49.0±3.1 | 50.9±0.0 | 71.9±2.5 | 55.0±9.6 | 74.4±4.7 |
| KDDCup99 | 62.5 | **99.2** | 98.9±0.1 | 55.9±24.3 | 50.4±36.2 | 72.8±16.1 | 76.2±14.4 | 47.4±1.6 | 50.8±0.0 | 65.1±2.8 | 64.0±9.3 | 98.9±0.1 |
| SpamBase | 45.1 | 64.4 | 62.1±2.0 | 50.5±13.6 | 52.3±35.7 | 49.1±6.6 | 39.1±1.9 | 33.9±3.0 | 49.2±0.0 | 57.5±2.4 | / | **68.4**±1.0 |
| optdigits | 58.8 | 61.5 | 71.0±4.7 | 52.2±23.6 | 56.8±29.1 | **89.1**±6.8 | 55.0±4.4 | 42.4±12.0 | 48.6±0.2 | 89.0±1.2 | 79.7±8.7 | 76.2±5.5 |
| mnist | 64.5 | 83.8 | 79.8±1.8 | 53.7±12.6 | 56.2±32.9 | 72.6±0.9 | **88.4**±1.3 | 49.4±0.3 | 49.2±0.1 | 86.5±0.6 | 55.8±6.9 | 86.7±2.0 |
| musk | 41.2 | 95.5 | **99.9**±0.1 | 68.4±20.3 | 54.2±36.1 | 58.8±7.2 | 99.8±0.2 | 50.0±0.0 | 47.4±0.2 | 99.8±0.1 | 97.0±2.2 | 98.2±0.7 |
| Arrhythmia | 72.6 | **77.4** | 75.0±1.3 | 61.4±5.2 | 48.0±28.0 | 51.5±3.6 | 73.6±0.9 | 34.2±3.3 | 48.1±0.4 | 74.4±1.0 | 37.8±2.8 | 74.0±0.5 |
| speech | 50.9 | 48.9 | 48.1±1.5 | 49.5±5.1 | 58.2±32.3 | 49.9±4.8 | 50.0±1.6 | 48.9±1.8 | 56.8±0.3 | 54.1±1.4 | 47.5±5.3 | **62.0**±1.8 |
| InternetAds | 65.5 | 67.7 | 68.8±2.0 | 70.3±3.6 | 49.0±36.4 | **88.0**±5.7 | 67.2±2.8 | 38.1±5.4 | 51.3±0.1 | 81.2±0.6 | / | 68.7±0.8 |
| AUC↑(rank↓) | 64.4(8) | 78.2(4) | 79.6(3) | 57.0(9) | 51.2(11) | 68.6(7) | 70.5(6) | 51.3(10) | 47.6(12) | 82.1(2) | 73.2(5) | **83.0(1)** |
| Rank_avg↓ | 7.0 | 4.7 | 4.4 | 8.7 | 9.4 | 6.2 | 6.4 | 9.8 | 9.9 | 2.7 | 6.1 | **2.5** |
| Rank_std↓ | 2.9 | 2.5 | 2.5 | 1.5 | 2.5 | 3.0 | 3.1 | 2.4 | 2.0 | 1.3 | 2.8 | **1.1** |

without domain-specific knowledge. REPEN uses representation-learning techniques to map high-dimensional data into low-dimensional embeddings and can be complementary to NOD.

## 4.3 ANOMALY DETECTION OF IMAGES

We also evaluate NOD in image anomaly detection. Similar to N_AD, the pre-trained ResNet152 (He et al., 2016) model is used to reduce the dimension of image data from 3*32*32 to 1*2048. To avoid possible "supervised" signals in the embeddings, only the test sets of CIFAR10 are used, with 950 samples from one class as inliers and 50 outliers randomly selected from the other nine.

| normal class | D_SVDD | DROCC | N_AD | REPEN | NOD |
|---|---|---|---|---|---|
| airplane | 60.7±0.5 | 87.3±4.3 | 91.2±0.1 | 87.1±0.6 | **95.3**±0.1 |
| automobile | 60.0±1.0 | 93.8±1.4 | **97.0**±0.1 | 86.2±4.0 | 96.6±0.1 |
| bird | 48.4±0.5 | 79.8±2.2 | 85.7±0.4 | 71.2±1.5 | **88.0**±0.2 |
| cat | 57.6±0.8 | 78.3±1.3 | 86.4±0.2 | 78.1±1.3 | **90.0**±0.1 |
| deer | 56.8±0.8 | 80.2±0.8 | 90.8±0.2 | 91.2±1.4 | **95.6**±0.1 |
| dog | 63.3±0.9 | 83.5±2.5 | 92.0±0.3 | 77.8±1.6 | **92.3**±0.3 |
| frog | 59.0±1.4 | 91.7±4.9 | 91.5±0.5 | 84.0±1.2 | **97.0**±0.1 |
| horse | 59.9±1.7 | 89.1±3.1 | **96.5**±0.1 | 85.1±1.8 | 95.2±0.1 |
| ship | 75.9±1.8 | 91.8±0.9 | 95.4±0.0 | 89.6±1.1 | **97.0**±0.1 |
| truck | 67.2±0.7 | 90.0±4.1 | 96.8±0.1 | 90.5±1.1 | **97.7**±0.0 |
| AUC_avg↑ | 60.9±1.0 | 86.5±2.5 | 92.3±0.2 | 84.1±1.6 | **94.4**±0.1 |

Table 2: ROC_AUC (%) on image data.

**bird** **truck**

inlier outlier

Figure 3: Embedding visualization by t-SNE.

Table 2 shows the performance of four strong baseline detectors and NOD. NOD demonstrates superior and consistent performance in nearly all datasets. Additionally, we observe that all detectors have almost identical trends across the ten datasets, albeit with different values: showing good performance in *ship* and *truck* while showing poorer performance in *bird* and *cat*. To understand the underlying reasons for this observed trend, we use t-SNE to visualize *bird* and *truck* as shown in Fig.3. Observing the relative density of inliers and outliers, it is evident that the outliers in *birds* pose a more significant violation of Assumption 1 compared to those in *truck* (highlighted by the red circle in *bird*). The underlying density patterns of the inliers and outliers in the data have a significant impact on the performance of all approaches, including NOD. We also observe that embeddings obtained from different pretraining models can have a notable influence on UOD performance (Appendix H).

## 4.4 IMPACTS OF DIFFERENT TYPES OF NOISE

In this section, we test the effectiveness of different negative sampling methods. In addition to "Uniform Random" (UR) and "Fast Poisson disk sampling" (FPD), the "Subspace" in LUNAR and

"GAN" in SO-GAAL are used to generate negative samples. Fig. 4 shows the results of different negative sampling methods. The subspace method (LUNAR) generates negative samples (noise

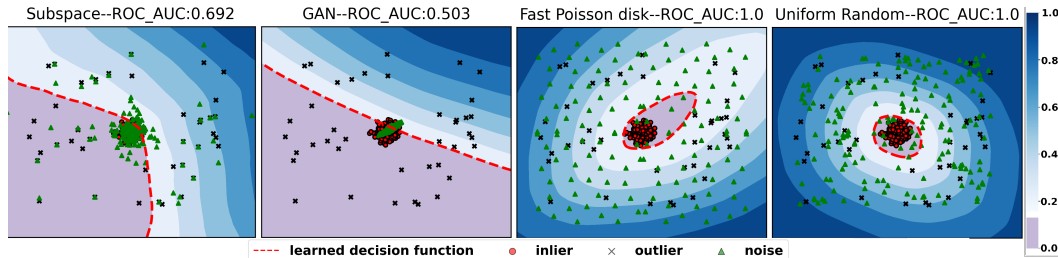

Figure 4: Comparison of different negative samples.

points) around positive ones (including inliers and outliers) with specific perturbations. These negative samples can hardly cover the whole space. Similarly, noise points generated from GAN tend to be located close to the inliers. However, due to the lack of labels, SO-GAAL and LUNAR find it difficult to control the distance between negative and positive samples, resulting in frequent model collapses. In comparison, negative samples generated from both FPD and UR are almost uniformly distributed in the entire $\mathbb{R}^{dim}$ space and serve as reliable reference points for a classifier for learning. Their anomaly scores gradually increase as the points move away from the cluster of inliers. Appendix G also illustrates results from data with multiple clusters.

## 4.5 FITTING WITH DIFFERENT CLASSIFIERS

According to Sec. 3, one of the preconditions of NOD is that the classifier obeys the *smoothness prior*. However, not all classifiers are designed with this assumption. We analyze the performance differences in applying LightGBM, SGD and Adam on a synthetic 2D dataset. The results for more classifiers are in Appendix F. Fig. 5 visualizes the anomaly score distributions in the 200th and

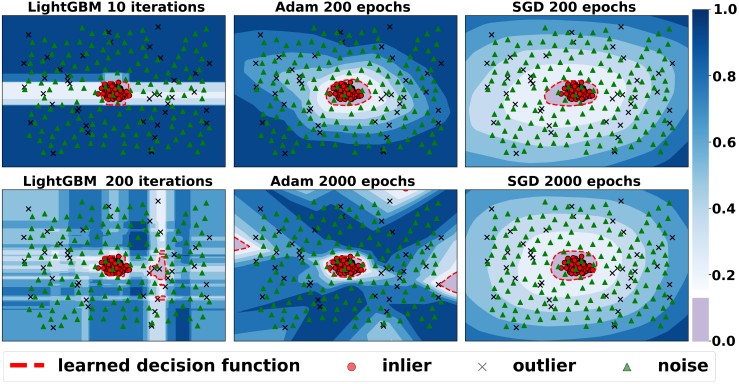

Figure 5: Score distributions in 2D data under different rounds

2000th epochs (for LigthGBM, the 10th and 200th). SGD has almost two identical score distributions between the 200th and 2000th epochs, while Adam overfits in the 2000th epoch. LightGBM always generates very rigid boundaries between positive samples and noises. It can hardly find a smooth distribution estimation with its rigorous separation strategy, even at the beginning of training.

## 4.6 DESIGN ANALYSIS

This section examines the effects of other designs in NOD, including the ratios of negative samples, and the usage of earlystop, including with no earlystop or earlystop with a validation set by randomly selected 15% samples from both samples and noises. X-axis values denote the dataset index ordered as Table 1. $|X|$ is normalized to 1 for the 22 datasets according to the performance of NOD. Dots above 1 indicate improved performance, while those below 1 indicate underperformance.

Fig. 6(a) shows the relative ROC_AUC ratios (Y-axis) with different ratios of negative samples. We observe that the performance generally deteriorates when there are too many / little negative samples. Fig. 6(b) shows the performance of our curve-based earlystop and validation-based earlystop. For most datasets, no earlystop might introduce possible overfitting, which results in performance degradation. The traditional validation-based earlystop also has inferior performance as the sampling for validation sets might change the distribution of the data, especially for small-size datasets.

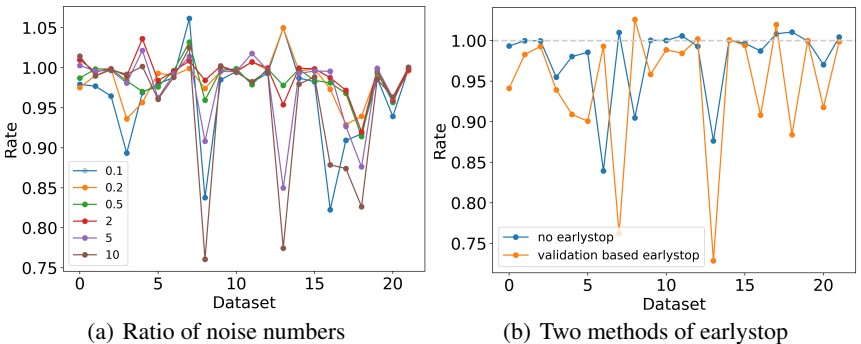

(a) Ratio of noise numbers          (b) Two methods of earlystop

Figure 6: Performance under different designs. X axis is the datasets, in the same order as Table 1
.

## 4.7 SCALABILITY ANALYSIS

Table 3 shows the time and space complexity of different detectors in two real-world datasets and three synthetic datasets. Here, $10^5 - 10^2$ denotes a data set of 100,000 samples and 100 features. The baselines are performed on a server with EPYC 7552*2 with 512G memory and V100 32G. The number of parallel execution cores for LOF, IForest, and ECOD is set to 50. For DNN-based solutions, both training and prediction time are counted. The results show that LOF suffers from computational scalability, while IForest exhibits high space complexity, OOM in $10^6$-$10^3$. LUNAR needs to maintain a graph with all samples and encounters OOM-G. In contrast, NOD is efficient in massive datasets, with its time complexity nearly linear with the sample size. This advantage is of paramount importance in many real-world applications.

Table 3: Time in seconds for different scales of datasets, – means the result is not obtained within 2 hours. OOM denotes the out-of-memory errors with 512G memory; OOM-G is on V100 32G.

| Dataset | LOF | IForest | ECOD | LUNAR | SO-GAAL | NOD |
|---------|-----|---------|------|-------|---------|-----|
| InternetAds | 2.58 | 3.30 | 11.59 | 4.79 | 4.30 | 0.54 |
| KDDCup99 | 299.69 | 4.40 | 13.76 | 4.56 | 2.37 | 0.44 |
| $10^5$-$10^2$ | 897.79 | 13.76 | 17.52 | 11.60 | 10.81 | 0.97 |
| $10^5$-$10^3$ | 1163.87 | 85.83 | 58.62 | OOM-G | 33.90 | 12.62 |
| $10^6$-$10^3$ | – | OOM | 687.74 | OOM-G | 103.40 | 24.56 |

## 5 CONCLUSION

This paper proposes a simple, effective, scalable, unsupervised outlier detection solution called NOD. NOD converts the original UOD problem into a density estimation problem and solves it with a simple MLP and standard back-propagation mechanism with uniform noise as guidelines. We theoretically prove that the trained classifier can effectively separate outliers if the outlier is not as dense as the inliers. In 22 real-world datasets, NOD achieves the best outlier detection performance in average ROC_AUC, outperforming 11 strong outlier detectors. Further experiments show that uniform noise and MLP designs contribute to the superior performance of NOD. We also experimentally demonstrate that NOD is not sensitive to its hyperparameters. For further work, we plan to improve NOD's performance on datasets with small but dense outlier clusters, violating the assumption that outliers are more sparsely distributed than inliers.

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

# Noise-guided Unsupervised Outlier Detection Appendix

**Anonymous authors**

## A  Theoretical Analysis

For the binary classification problem, we use the binary cross entropy loss to optimize the classifier $f(x)$ by minimizing the loss $\mathcal{L}_f^n$:

$$\mathcal{L}_f^n = -\Big(\sum_{i=0}^{|X|} \log(1 - f(x_i)) + \sum_{k=0}^{|X^-|} \log f(x_k)\Big). \tag{1}$$

When we have arbitrarily large samples, the weak law of large numbers shows that the objective function $\mathcal{L}_f^n$ converges in probability to $\mathcal{L}_f$:

$$\mathcal{L}_f = -\big(E_X(\log(1 - f(x))) + E_{X^-}(\log f(x^-))\big). \tag{2}$$

Let $p(x, y) = p(y)p(x|y)$ be an expanded generative model for x defined as:

$$\begin{aligned}
x &\sim a(x) \quad if \quad y = 0, \\
x &\sim b(x) \quad if \quad y = 1
\end{aligned} \tag{3}$$

When the number of positive and negative samples is equal, we can express the loss function as:

$$\mathcal{L}_f = -\int \big(\log(1 - f)a(x) + \log(f)b(x)\big)dx. \tag{4}$$

$$\frac{\partial \mathcal{L}_f}{\partial f} = -\int \Big(\frac{1}{f-1}a(x) + \frac{1}{f}b(x)\Big)dx. \tag{5}$$

When the derivative is constantly zero, the objective function achieves an extremum. By doing this, we can obtain an optimized classifier:

$$f^* \approx \frac{b(x)}{a(x) + b(x)} = p(y = 1|x) \tag{6}$$

$f(x)$ is the output of the classifier with input $x$ and is the predicted anomaly score of the sample $x$. We can obtain the optimal classifier $f^*(x) \approx p(y = 1|x)$ after minimizing the loss $\mathcal{L}_f$. The proof above referenced the counterparts from (Gutmann & Hyvärinen, 2012).

To distinguish inliers and outliers with limited samples, two restrictions are placed, one assumption on the datasets and another on the optimizer. We then provide a simplified proof of the correctness of NOD.

**Assumption 1. [Distribution assumption]** *Outliers are sparser distributed than inliers and should be sufficiently distant from any inlier.*

Due to the highly unbalanced nature of the sample, we assume that outliers are sparser distributed than inliers and nonoverlapping with inliers. Without this assumption, it would be very hard to differentiate between inliers and outliers. This assumption has been adopted in density-based studies. However, the difficulty lies in how to effectively and efficiently estimate the density of high-dimensional data, due to the "curse of dimensionality". Many UOD calibrate the anomaly score based on the localized distance/density estimation to reduce computation cost. It is difficult for them to use samples beyond their scope. Fig. 1(a) of a toy sample shows the limitations of kNN in the localized calculation.

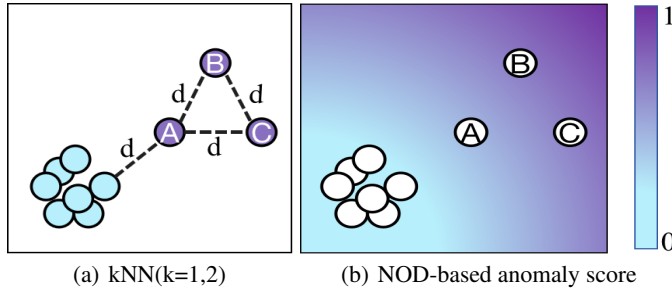

(a) kNN(k=1,2)      (b) NOD-based anomaly score

Figure 1: Anomaly score with kNN and NOD on the 2D dataset. Sample A is close to the normal cluster and with lower score than B,C. However, kNN(k=1,2) assign same socre to A,B and C.

Certainly, the sparse assumption may not be valid for outliers that are clustered together. However, we argue that we can only effectively address these small clusters with domain-specific knowledge. Section 4.3 in paper contains a discussion on the effects of clustered outliers.

This *Smoothness Prior* (Rosca et al., 2020) specifies that the changing rate of $f(x)$ across the whole value space is below a certain threshold and has been wildly used in designing many optimizers, especially those optimizers used in DNNs to estimate a smooth function, e.g., Adam (Kingma & Ba, 2014) and SGD (Bottou & Bousquet, 2007).

**Lemma 1.** When the value space is limited, using a limited amount of uniform noise, it is ensured that $\rho(x_i) > \rho(x_k) > \rho(x_j)$, where $x_i \in X_n, x_j \in X_o, x_k \in X^-$, and $\rho(\cdot)$ is the density function.

**Proof.** For $\forall x_{j1}, x_{j2} \in X_o, \forall x_{i1}, x_{i2} \in X_n$ we have $d(x_{j1}, x_{j2}) > 4\sqrt{dim}d(x_{i1}, x_{i2})$ where $dim$ is the space dimension for the dataset and the 4 is a scaling factor. We let $D = \max_{i1,i2}d(x_{i1}, x_{i2})$, $S$ be the dataset space, and $\rho(\cdot)$ be the density function. $\rho(x) = \max_y \frac{C(N(x,d(x,y)),x)}{Sqr(N(x,d(x,y)))}$, where x,y come from the same dataset, $N(x, d_x) = \{z|d(x,z) \leq d_x, z \in S\}$, $C(N, x)$ means the number of the data which has the same tag as $x$ and is in the subset $N$, $Sqr(N)$ means the volume of the subset $N$. Construct the noise following a uniform distribution, in which the distance between two adjacent points is $4D$; we have: $\min_{j1,k1}d(x_{j1}, x_{k1}) < \frac{\sqrt{dim}}{2} * 4D < 4\sqrt{dim}D < \min_{j1,j2}d(x_{j1}, x_{j2})$,

where $x_{j1}, x_{j2} \in X_o, x_{k1} \in X^-$. This indicates that noise is distributed near the outlier instead of the outlier and $\forall x_j \in X_o, \forall x_k \in X^-, \rho(x_k) > \rho(x_j)$. Given that $\max_{x \in S}\min_{x_{k1} \in X^-} d(x, x_{k1}) = 2\sqrt{dim}D$, we can generate the uniform noise data to guarantee $\exists d(x_{i1}, x_{k1}) \geq 2D$. Then:

$$\min_{i1,k1}d(x_{i1}, x_{k1}) > 2D - \max_{i1,i2}d(x_{i1}, x_{i2}) = \max_{i1,i2}d(x_{i1}, x_{i2}).$$

It means that $\forall x_i \in X_n, \forall x_k \in X^-, \rho(x_k) < \rho(x_i)$. Thus, $\rho(x_j) < \rho(x_k) < \rho(x_i)$. In other words, it is always possible to generate a uniform noise that has a density between that of the inliers and outliers.

**Lemma 2.** Let $D = \max_j \big( \min_k d(x_j, x_k)\big)$, where $x_j \in X_o, x_k \in X^-$. There exists an optimized $f^*(x)$ with respect to Equ. 1 that satisfy $\forall x_j \in X_o, f^*(x_j) \geq 1 - MD, MD < 1$.

This lemma shows that the learned optimized function always gives the outlier an anomaly score bigger than a certain positive value.

**Proof.[proof by contradiction]** If $\exists x_j \in X_o$, s.t. $f(x_j) < 1 - MD$, with smoothness prior (i.e., $x_j, x_k$ are in a subspace with $d(x_j, x_k) < \epsilon$, thus, $f(x_j) \to f(x_k)$), $\exists x_k \in X^-, f(x_k) < 1 - MD + Md(x_j, x_{k*}) < 1$. $x_{k*}$ is the closest of $x_k \in X^-$ to $x_j$. Thus, there is an optimal classification value $f^*(x)$, so that $f(x_k) \leq f^*(x_k), \forall x_k \in X^-$, and we further define $f^*(x)$ as:

$$f^*(x) = \begin{cases} 1, & \forall x \in X^-, \\ f(x), & \forall x \in X_n, \\ 1 - MD_x, & \forall x \in X_o. \end{cases} \tag{7}$$

Here, $D_x = \min_k d(x_k, x), x_k \in X^-$. According to Equ. 1, we define $C$ as a sample set that belongs to the same subspace as $x_j$ and $\forall x \in C, x \in X^-$. Let $g(x) = -\big(\log\big(1 - (x - \delta)\big) + \log(x)\big), 0 < \delta < 1$, because $g(1) < g(x), \forall \delta \leq x < 1$. With Assumption 1 and Lemma 1, one outlier has little effect on another outlier. Thus, we can only care about one outlier and noise samples around that outlier. We use $\mathcal{L}_{f(x_j)}$ to represent the loss function, where $x_j$ is the outlier:

$$
\begin{aligned}
\mathcal{L}_{f(x_j)} &= -\Big( \sum_k^{|C|} \log f(x_k) + \log(1 - f(x_j)) \Big) \\
&\geq -\Big( \log(1 - f(x_j)) + \log f(x_{k*}) + \sum_k^{|C| \setminus \{x_{k*}\}} \log f^*(x_k) \Big) \\
&\geq -\Big( \log(1 - f^*(x_j)) + \log f^*(x_{k*}) + \sum_k^{|C| \setminus \{x_{k*}\}} \log f^*(x_k) \Big) \\
&= \mathcal{L}_{f^*(x_j)},
\end{aligned}
\tag{8}
$$

where $f^*(x_k) = 1, f^*(x_j) = 1 - \min_k Md(x_j, x_k), x_k \in X^-, x_j \in X_o$. Therefore, if there exists $\exists x_j \in X_o$, s.t. $f(x_j) < 1 - MD$, it is theoretically possible to find $f^{(}x_j)$ that minimizes the loss, which contradicts the fact that f (x) is optimal. Thus Lemma 2 holds true.

**Theorem 1.** Each predicted value of the outlier is higher than each predicted value of the inlier. $\forall x_i \in X_n, x_j \in X_o$, it holds that $f^*(x_j) > \lambda > f^*(x_i)$, where $\lambda$ is a boundary value.

**Proof.** Due to the high density of inliers, $\forall x_i \in X_n$, when $\rho(x_i) \to +\infty$, we have $f^*(x_i) \to 0$. Thus, there exists a density value $\rho_0$, s.t. $\forall x_i \in X_n$, we have $f^*(x_i) < \tau$. According to *Lemma 2*, it is possible to learn a classifier $f^*(\cdot)$ that satisfies the following conditions: $\forall x_i \in X_n, x_j \in X_o, f^*(x_i) < \tau \leq \lambda \leq 1 - MD \leq f^*(x_j)$. For instance, when $M < \frac{1}{2D}, \tau = 0.5$. Therefore, *Theorem 1* holds.

This theorem establishes that the anomaly scores of outliers are higher than those of inliers. If we have an outlier ratio, the classifier $f^*(x)$ can distinguish between $X_n$ and $X_o$. Fig. 1(b) shows the anomaly score distribution in the toy example. It clearly shows that NOD can effectively balance the impact of both local and remote samples with the support of uniform noise. The anomaly scores span the entire value space and exhibit a gradual increase as the points move farther from the inlier center. Therefore, NOD can identify the anomaly degree of samples A, B, and C.

## B    SUMMARY OF 22 REAL-WORLD DATASETS

Table 1 summarizes 22 real-world datasets used for evaluating UOD. (mat) represents dataset from ODDS[1] and (arff) from DAMI[2]. These datasets are highly representative in terms of diversity in feature dimensions, data volume, and anomaly proportions. The following experiments are the average results obtained from 20 independent experiments on these 22 datasets.

## C    PERFORMANCE ON 22 DATASETS USING 22 OUTLIER DETECTORS

NOD is compared with 21 other outlier detectors, including classical methods: kNN(Ramaswamy et al., 2000), LOF(Breunig et al., 2000), HBOS(Goldstein & Dengel, 2012), OC-SVM (Schölkopf et al., 2001), COPOD(Li et al., 2020), ECOD(Li et al., 2022), IForest(Liu et al., 2008), SUOD(Zhao et al., 2021), LSCP(Zhao et al., 2019a) and DNN-based detectors: Deep SVDD (D_SVDD) (Ruff et al., 2018), AE(Xia et al., 2015), VAE (Kingma & Welling, 2013), LUNAR (Goodge et al., 2021), DROCC (Goyal et al., 2020), GOAD (Bergman & Hoshen, 2020), Neutral AD (N_AD) (Qiu et al., 2021), SO-GAAL (Liu et al., 2020), REPEN(Pang et al., 2018), DAGMM(Zong et al., 2018), ICL(Shenkar & Wolf, 2021) and flows_ood(Kirichenko et al., 2020).

---

[1] http://odds.cs.stonybrook.edu
[2] http://www.dbs.ifi.lmu.de/research/outlier-evaluation/DAMI

Table 1: Summary of 22 real-world datasets (Ratio means Outlier Ratio).

| Dataset | $dim$ | Sample | Ratio (%) | Dataset | $dim$ | Sample | Ratio (%) |
|---|---|---|---|---|---|---|---|
| pima(mat) | 8 | 768 | 34.90 | breastw(mat) | 9 | 683 | 34.99 |
| WBC(arff) | 9 | 223 | 4.48 | wine(mat) | 13 | 129 | 7.75 |
| HeartDisease(arff) | 13 | 270 | 44.44 | pendigits(mat) | 16 | 6870 | 2.27 |
| Lymphography(arff) | 18 | 148 | 4.05 | Hepatitis(arff) | 19 | 80 | 16.25 |
| Waveform(arff) | 21 | 3443 | 2.90 | wbc(mat) | 30 | 378 | 5.56 |
| WDBC(arff) | 30 | 367 | 2.72 | WPBC(arff) | 33 | 198 | 23.74 |
| satimage-2(mat) | 36 | 5803 | 1.22 | satellite(mat) | 36 | 6435 | 31.64 |
| KDDCup99(arff) | 41 | 60839 | 0.40 | SpamBase(arff) | 57 | 4207 | 39.91 |
| optdigits(mat) | 64 | 5216 | 2.88 | mnist(mat) | 100 | 7603 | 9.21 |
| musk(mat) | 166 | 3062 | 3.17 | Arrhythmia(arff) | 259 | 450 | 45.78 |
| speech(mat) | 400 | 3686 | 1.65 | InternetAds(arff) | 1555 | 1966 | 18.72 |

For kNN, LOF, HBOS, OC-SVM, COPOD, ECOD, IForest, SUOD, LSCP, AE, VAE, and D_SVDD, we use the implementations from PyOD (Zhao et al., 2019b) which is a popular and open-source Python library for Outlier Detection. For others, we use the code given in their papers. In particular, from their source code, D_SVDD, DROCC, GOAD, N_AD, and LUNAR demand pure inliers, i.e., these methods select inliers based on labels and use only inliers as training data. For a fair comparison, we adapt them to the UOD setting by using the original dataset containing both inliers and outliers for model training. The comparison of experimental results using the initial settings of the paper and the UOD settings is presented in Sec. D.

**Detailed Hyperparameter Settings.** For kNN, LOF, HBOS, OC-SVM, COPOD, ECOD, IForest, SUOD and LSCP, we use default settings in the PyOD library where n_neighbors is 5 in kNN, n_bins is 10 in HBOS, n_neighbors is 20 in LOF and OC-SVM uses the sigmoid kernel. SUOD and LSCP are ensemble learning methods, and their basic detector composition is [LOF, LOF, LOF, LOF, COPOD, IForest, IForest], the parameters n_neighbors for the first four LOF algorithms are 15, 20, 25 and 35 respectively.

For DNN models, AE, VAE and D_SVDD use the sigmoid activation function and the SGD optimizer. We train them using 500 epochs with a learning rate of 0.005 and 2 hidden layers. The hidden layer dimensions are $\frac{dim}{2}$ and $\frac{dim}{4}$ for the two models, respectively. We train DROCC 100 epochs where 50 epochs are with CELOSS. The number of hidden nodes for the LSTM model is 128, and the SGD optimizer is used with a 0.005 learning rate and 0.99 momentum. We use the config file "config_arrhy.yml" provided in the source code from N_AD paper where *residual* transformation, Adam with 0.005 learning rate, 5 hidden layers with 64 dimensions and DCL loss are used. For SO-GAAL, the SGD optimizer is used with a 0.0001 learning rate for the generator and a 0.01 learning rate for the discriminator. LUNAR uses kNN with 20 n_neighbors to build a graph and constructs a discriminator with 4 layers with the Tanh activation function. SGD optimizer with a learning rate of 0.01 is adopted. REPEN uses the "rankod.py" from the paper code library to evaluate and the number of training epochs is 50. DAGMM adopts the Adam optimizer with a learning rate of $10^{-4}$ and a training epochs count of 200, where the *gmm_k* parameter is 4. flows_ood uses Adam optimizer with $10^{-3}$ learning rate and $5 \times 10^{-5}$ L2 regularization weight decay, and the number of training epochs is 100. For flows_ood, we use the file "train_unsup_ood_uci.py" from the paper code library to train.

Tables 2, 4 and Tables 3, 5 show the results of NOD compared to other outlier detectors, in terms of AUC_ROC, F1-score respectively. Table 2 and Table 3 show the performance comparison between NOD and traditional methods, while Table 4 and Table 5 show the comparison with deep learning methods The results show that most outlier detectors display significant performance variance on different datasets. Original data distribution highly influences the performances of traditional outlier detection algorithms due to their strong data assumptions, and only a tiny fraction of them can achieve good performance on the 22 datasets. For instance, kNN performs well on the *wine(mat)*, *wbc(mat)*, *breastw(mat)*, *Lymphography(arff)*, *WBC(arff)*, *satimage-2(mat)* and *WDBC(arff)* datasets, but poorly on others as its performance is highly influenced by $k$.

Table 2: Results in ROC_AUC (%) of all 9 compared classical detectors (average of 20 independent trials).

| Dataset | kNN | LOF | HBOS | OC-SVM | COPOD | ECOD | IForest | SUOD | LSCP | nod |
|---------|-----|-----|------|--------|-------|------|---------|------|------|-----|
| pima | 60.76 | 53.84 | 68.6 | 50 | 65.4 | 51.73 | 67.33±0.9 | 64.93±0.76 | 61.7±0.81 | 62.9±2.95 |
| breastw | 97.53 | 38.32 | 98.5 | 0.49 | 99.44 | 99.14 | 98.7±0.16 | 90.95±1.76 | 75.69±1.56 | 99.29±0.24 |
| WBC | 98.73 | 83 | 98.2 | 0.75 | 99.06 | 99.01 | 99.04±0.21 | 98.31±0.31 | 97.4±0.12 | 99.16±0.21 |
| wine | 99.62 | 99.75 | 76.6 | 50 | 86.72 | 71.01 | 79.23±3.7 | 98.5±0.19 | 97.98±0.44 | 97.21±1.16 |
| HeartDisease | 60.53 | 50.05 | 74.7 | 14.78 | 69.46 | 58.81 | 62.22±1.24 | 61.77±1.01 | 57.02±0.84 | 67.08±3.76 |
| pendigits | 70.87 | 47.94 | 92.8 | 76.67 | 90.48 | 90.9 | 94.41±1.1 | 86.96±1.13 | 69.57±3 | 91.59±1.84 |
| Lymphography | 99.65 | 97.65 | 99.8 | 8.1 | 99.65 | 99.53 | 99.91±0.08 | 99.54±0.15 | 98.17±0.66 | 99.74±0.29 |
| Hepatitis | 66.88 | 62.57 | 77.7 | 69.8 | 80.37 | 78.65 | 69.41±1.89 | 73.08±2.56 | 72.24±1.78 | 69.69±3.39 |
| Waveform | 73.7 | 73.41 | 70 | 61.03 | 73.43 | 72.03 | 70.79±1.86 | 75.2±1.56 | 74.95±0.8 | 80.74±4.8 |
| wbc | 95 | 92.97 | 95.8 | 1.56 | 96.36 | 90.01 | 93.7±0.81 | 95.06±0.5 | 94.41±0.5 | 95.9±0.85 |
| WDBC | 99.41 | 98.15 | 93.1 | 50 | 97.09 | 91.74 | 93.53±0.91 | 96.77±0.22 | 95.77±0.15 | 97.39±0.47 |
| WPBC | 51.54 | 51.85 | 54.5 | 44.86 | 52.33 | 48.01 | 49±1.52 | 50.89±0.52 | 49.94±0.92 | 57.76±1.34 |
| satimage-2 | 92.96 | 53.25 | 97.2 | 50 | 97.45 | 97.32 | 99.36±0.1 | 98.45±0.1 | 90.04±3.19 | 99.51±0.1 |
| satellite | 67.01 | 53.95 | 76.6 | 50 | 63.35 | 74.63 | 70.75±1.67 | 69.87±0.44 | 61.46±0.52 | 74.43±4.66 |
| KDDCup99 | 43.9 | 62.54 | 98.4 | 91.33 | 99.19 | 99.24 | 98.91±0.08 | 99.03±0.05 | 93.73±1.46 | 98.94±0.13 |
| SpamBase | 48.64 | 45.13 | 63.7 | 30.43 | 67.71 | 64.45 | 62.1±1.96 | 61.16±0.85 | 55.97±0.96 | 68.4±0.98 |
| optdigits | 43.57 | 58.79 | 87 | 53.6 | 68.24 | 61.53 | 70.97±4.69 | 68.5±1.23 | 60.65±1.87 | 76.15±5.54 |
| mnist | 79.41 | 64.49 | 68.7 | 91.09 | 77.39 | 83.81 | 79.8±1.8 | 80.26±0.61 | 72.15±0.61 | 86.3±2.04 |
| musk | 30.38 | 41.24 | 99.8 | 1.1 | 94.63 | 95.5 | 99.97±0.05 | 91.71±0.81 | 67.39±9.94 | 98.18±0.69 |
| Arrhythmia | 74.33 | 72.59 | 74.8 | 66.83 | 75.76 | 77.37 | 75.05±1.3 | 75.22±0.28 | 73.29±0.33 | 73.98±0.54 |
| speech | 49.29 | 50.87 | 47.6 | 50.57 | 49.11 | 48.9 | 48.12±1.53 | 49.3±0.58 | 50.15±0.21 | 62.02±1.78 |
| InternetAds | 71.27 | 65.54 | 68.3 | 38.35 | 67.64 | 67.67 | 68.81±2 | 74.62±0.83 | 71.92±2.13 | 68.71±0.75 |
| AUC_avg | 71.59 | 64.45 | 81.0 | 43.24 | 80.47 | 78.23 | 79.6±1.3 | 80.0±0.7 | 74.6±1.5 | 83.0±1.8 |

Table 3: Results in F1-score (%) of all 9 compared classical detectors (average of 20 independent trials).

| Dataset | kNN | LOF | HBOS | OC-SVM | COPOD | ECOD | IForest | SUOD | LSCP | nod |
|---------|-----|-----|------|--------|-------|------|---------|------|------|-----|
| pima | 44.8 | 34.11 | 50.75 | 0 | 48.88 | 37.31 | 51.38±1.27 | 47.82±1.42 | 44.13±1.19 | 48.99±1.81 |
| breastw | 87.88 | 13.84 | 93.5 | 0 | 94.56 | 92.89 | 92.33±0.63 | 78.01±3.1 | 51.26±3.35 | 94.46±0.9 |
| WBC | 70.59 | 0 | 70 | 0 | 80 | 80 | 70±3.16 | 63.61±6.7 | 59.85±2.35 | 75.5±4.97 |
| wine | 77.78 | 66.67 | 0 | 0 | 40 | 20 | 14.5±6.69 | 70.49±1.53 | 68.76±7.2 | 67±11 |
| HeartDisease | 44.34 | 45.3 | 70 | 15.83 | 60.83 | 52.5 | 51.5±1.25 | 51.68±1.6 | 45.75±0.94 | 59.29±3.59 |
| pendigits | 7.25 | 6.36 | 32.05 | 16.03 | 26.28 | 25 | 32.76±3.79 | 10.7±1.18 | 15.4±3.3 | 20.06±7.69 |
| Lymphography | 83.33 | 72.73 | 83.33 | 0 | 83.33 | 83.33 | 90±8.16 | 84.23±2.69 | 69.46±6.08 | 85±10.41 |
| Hepatitis | 0 | 17.39 | 30.77 | 30.77 | 46.15 | 38.46 | 19.23±3.85 | 22.01±5.32 | 21.23±4.53 | 22.69±7.08 |
| Waveform | 19.65 | 12.09 | 7 | 6 | 4 | 8 | 7.1±1.48 | 6.12±1.3 | 16.94±1.08 | 12.5±4.79 |
| wbc | 45 | 43.24 | 61.9 | 0 | 71.43 | 42.86 | 53.57±5.4 | 56.24±5.01 | 57.53±3.48 | 65±4.83 |
| WDBC | 80 | 84.21 | 40 | 0 | 80 | 50 | 64±4.9 | 79.43±1.36 | 62.67±4.39 | 63.5±4.77 |
| WPBC | 13.64 | 19.15 | 19.15 | 14.89 | 21.28 | 14.89 | 14.79±1.84 | 15.53±2.11 | 15.67±1.75 | 30.32±2 |
| satimage-2 | 40 | 4.92 | 64.79 | 0 | 74.65 | 63.38 | 87.75±2.14 | 31.99±3.03 | 36.52±4 | 89.15±1.42 |
| satellite | 49.46 | 36.22 | 56.83 | 0 | 48.04 | 55.16 | 57.59±1.49 | 56.24±0.64 | 44.55±0.7 | 49.7±7.64 |
| KDDCup99 | 7.74 | 0 | 39.02 | 53.66 | 45.93 | 45.53 | 40.92±1.35 | 37.29±6.96 | 30.7±6.67 | 37.36±1.5 |
| SpamBase | 40.04 | 34.26 | 51.53 | 23.94 | 56.46 | 54.14 | 50.21±2 | 50.41±0.81 | 43.76±1.28 | 57.33±0.99 |
| optdigits | 3.76 | 11.43 | 18.67 | 10.92 | 1.33 | 1.33 | 2.53±1.19 | 7.06±0.96 | 9.49±0.39 | 5.7±1.67 |
| mnist | 37.6 | 22.63 | 17.14 | 56.71 | 23.57 | 34.86 | 29.84±2.17 | 31.49±1.14 | 27.77±0.63 | 39.79±5.29 |
| musk | 1.4 | 3.73 | 90.72 | 0 | 36.08 | 40.21 | 96.8±3.86 | 14.15±3.35 | 15.02±8.43 | 65.15±7.09 |
| Arrhythmia | 64.82 | 62.69 | 64.56 | 57.28 | 64.56 | 66.5 | 64.95±1.61 | 66.4±1.04 | 63.64±0.64 | 64.49±0.82 |
| speech | 1.79 | 2.38 | 3.28 | 3.28 | 3.28 | 3.28 | 3.36±1.68 | 2.6±0.82 | 4.07±0.81 | 1.89±1.4 |
| InternetAds | 32.56 | 39.07 | 46.47 | 9.51 | 44.57 | 44.57 | 43.24±2.82 | 50.67±1.45 | 48.45±2.28 | 46.37±1.28 |
| F1_avg | 38.79 | 28.75 | 45.97 | 13.58 | 47.96 | 43.37 | 47.2±2.85 | 42.46±2.43 | 38.75±2.97 | 50.06±4.22 |

DNN-based methods: AE, VAE, D_SVDD, DROCC, GOAD, N_AD, SO-GAAL, LUNAR, DAGMM and flows_ood are in a similar situation. In particular, LUNAR relies on the kNN method, SO-GAAL has no clear criteria for the distance between positive and negative samples and VAE is based on the assumption that the inliers can be decoded from the encoding space better than the outliers. Moreover, D_SVDD, DROCC, GOAD, N_AD, and LUNAR need to use pure normal samples for training, contrary to the unsupervised setting. Therefore, we use the original datasets containing both inliers and outliers rather than only containing inliers to train these models. The following experimental results (Sec. D) show that training data mixed with some noise samples hurt their model performance. Except for REPEN, REPEN uses representation learning techniques to map high-dimensional data into low-dimensional embeddings and can be complementary to NOD. One of our future directions is to integrate representation learning techniques into NOD. With the loose assumption, NOD has a rather stable performance and achieves excellent ROC_AUC on almost all the tested datasets. It is worth noting that NOD has 9 average ROC_AUC scores above 0.95 on 22 datasets and NOD performs best among DNN methods with large margins. The results verify the effectiveness and robustness of NOD.

## D  PERFORMANCE ON DIFFERENT TRAINING SETTINGS

Following the papers of DROCC, GOAD, N_AD and LUNAR, these approaches need pure normal samples (inliers) to train the model. Since we focus on the unsupervised domain, these models are trained using original data (including outliers) as training data. Table 6 shows comparative results using the original paper setting and unsupervised setting. The results show that GOAD, N_AD and LUNAR are interfered by the noise in the training data. On the contrary, DROCC generally performs better in the unsupervised setting. This is because DROCC can be extended to solve One-class Classification with Limited Negatives. For both versions, their results are inferior to NOD.

Table 4: Results in ROC_AUC (%) of all 11 compared DNN-based detectors (average of 20 independent trials, "r" indicates that the code provided by the paper cannot run on this dataset).

| Dataset | D.SVDD | AE | VAE | LUNAR | DROCC | GOAD | N_AD | SOGAAL | REPEN | DAGMM | ICL | flows_ood | nod |
|---|---|---|---|---|---|---|---|---|---|---|---|---|---|
| pima | 48.78±10.83 | 48.45±1.12 | 57.71±0.64 | 50.46±0.07 | 48.25±30.17 | 44.96±2.57 | 49.88±1.45 | 50.76±1.18 | 64.4±2.8 | 59.02±4.97 | 49.14±3.91 | 65.47±0.62 | 62.9±2.95 |
| breastw | 78.01±19.58 | 59.52±4.36 | 85.61±1.24 | 49.45±0.13 | 46.75±31.92 | 77.15±2.98 | 70.37±1.98 | 97.6±0.3 | 98.8±0.33 | 96.75±2.69 | 78.65±3.42 | 92.23±4.56 | 99.29±0.24 |
| WBC | 89.41±14.21 | 97.88±0.04 | 98.08±0.02 | 47.08±0.31 | 53.75±31.77 | 5.69±3.25 | 85.81±2.31 | 95.69±0.52 | 99.16±0.22 | 84.31±13.01 | 75.15±8.12 | 98.08±0.32 | 99.16±0.21 |
| wine | 42.33±28.2 | 57.18±4.04 | 74.94±0.64 | 30.02±0.59 | 47.9±31.61 | 71.56±20.32 | 79.27±4.55 | 51.13±1.25 | 99.87±0.08 | 95.51±9.23 | 50.62±20.44 | 65.13±2.98 | 97.21±1.16 |
| HeartDisease | 48.78±17.75 | 34.33±1.24 | 49.35±0.9 | 47.87±0.25 | 38.6±26.85 | 47.87±3.23 | 46.2±2.37 | 42.45±8.38 | 66.01±2.75 | 77.06±4.86 | 55.16±6.45 | 66.36±1.4 | 67.08±3.76 |
| pendigits | 49.32±25.64 | 93.3±0.39 | 93.69±0.06 | 56.39±0.09 | 44.7±30.36 | 20.14±12.56 | 78.64±4.94 | 66.23±9.7 | 97.69±0.34 | 91.7±3.65 | 57.6±8.3 | 81.68±3.97 | 91.59±1.84 |
| Lymphography | 54.57±25.17 | 99.75±0.04 | 99.68±0.05 | 25.28±1.4 | 56.35±31.16 | 21.4±13.69 | 82.95±9.31 | 94.89±8.69 | 99.13±0.51 | / | 93.57±4.9 | 97.72±0.4 | 99.74±0.29 |
| Hepatitis | 50.3±18.98 | 75.87±0.04 | 74.65±0.3 | 46.49±4.95 | 39.7±25.79 | 39.23±5.63 | 39.16±11.92 | 44.29±9.69 | 76.85±5.61 | 60.68±12.55 | 57.7±7.99 | 59.43±2.59 | 69.69±3.39 |
| Waveform | 54.38±17.43 | 65.44±0.85 | 65.52±0.19 | 49.49±0.07 | 53.85±35.58 | 44.17±2.16 | 76.12±2.1 | 33.78±3.02 | 78.01±4.71 | 60.77±12.11 | 53.89±1.54 | 66.83±0.85 | 80.74±4.8 |
| wbc | 64.13±26.84 | 82.32±0.64 | 91.56±0.13 | 42.57±0.31 | 45.6±31.03 | 14.98±3.44 | 85.71±2.63 | 12.3±6.72 | 95.81±0.48 | 94.35±3.54 | 81.71±3.86 | 94.66±0.3 | 95.9±0.85 |
| WDBC | 47.26±27.53 | 82.65±0.38 | 89.76±0.16 | 47.63±0.25 | 62.55±30.12 | 9.94±3.77 | 96.65±0.95 | 50.36±0.31 | 98.92±0.28 | 88.97±13.35 | 92.95±1.68 | 93.93±0.55 | 97.39±0.47 |
| WPBC | 49.49±6.64 | 42.81±0.27 | 46.43±0.15 | 47.81±0.2 | 54.15±36.59 | 51.14±2.47 | 43.86±3.32 | 50.15±3.98 | 52.37±2.2 | 55.62±5.83 | 53.72±3.63 | 50.11±1.11 | 57.76±1.34 |
| satimage-2 | 61.82±32.82 | 97.47±0.04 | 97.76±0 | 55.36±0.05 | 58.6±30.64 | 87.85±8.24 | 97.19±0.71 | 44.75±10.05 | 99.86±0.05 | 88.7±10.25 | 78.45±6.2 | 99.41±0.21 | 99.51±0.1 |
| satellite | 53.61±13 | 70.19±0.41 | 62.22±0.07 | 50.93±0.01 | 50.4±33.59 | 48.23±2.9 | 70.23±2.23 | 48.96±3.08 | 71.93±2.55 | 55.04±9.62 | 57.4±7.73 | 70.7±0.8 | 74.43±4.66 |
| KDDCup99 | 55.92±24.26 | 99.02±0 | 99.02±0 | 50.85±0.02 | 50.4±36.21 | 89.35±8.13 | 76.2±14.44 | 47.42±1.62 | 65.1±2.79 | 64.05±9.28 | 93.82±1.88 | / | 98.94±0.13 |
| SpamBase | 50.54±13.61 | 50.45±0.39 | 54.65±0.03 | 49.18±0.02 | 52.3±35.76 | 46.15±3.04 | 39.08±1.88 | 33.88±3.02 | 57.49±2.36 | 47.72±3.61 | 47.72±3.61 | 45.11±2.55 | 68.4±0.98 |
| optdigits | 52.21±23.56 | 44.43±0.98 | 50.43±0.09 | 48.57±0.15 | 56.8±29.07 | 58.33±13.18 | 55.03±4.41 | 42.44±11.99 | 89.01±1.22 | 79.71±8.73 | 66.01±3.58 | / | 76.15±5.54 |
| mnist | 53.68±12.62 | 89.1±0.08 | 85.76±0.01 | 49.19±0.1 | 56.2±32.95 | 44.97±7.91 | 88.39±1.31 | 49.36±0.32 | 86.51±0.65 | 55.83±6.9 | 90.63±0.16 | / | 86.3±2.04 |
| musk | 68.44±20.34 | 100±0 | 100±0 | 47.35±0.24 | 54.15±36.14 | 83.55±16.24 | 99.8±0.15 | 50±0 | 99.83±0.1 | 97.03±2.15 | 99.83±0.05 | 95.87±3.27 | 98.18±0.69 |
| Arrhythmia | 61.42±5.16 | 73.3±0.01 | 73.19±0.02 | 48.1±0.44 | 48.05±28.02 | 42.05±3.09 | 73.56±0.88 | 34.16±3.33 | 74.38±0.98 | 37.8±2.84 | 72.23±0.87 | 75.06±0.32 | 73.98±0.54 |
| speech | 49.48±5.14 | 46.99±0.02 | 46.91±0.01 | 56.78±0.26 | 58.15±32.29 | 51.92±3.66 | 49.97±1.59 | 48.88±1.77 | 54.09±1.38 | 47.48±5.3 | 54.02±2.03 | 1.08±0.14 | 62.02±1.78 |
| InternetAds | 70.32±3.64 | 61.4±0.01 | 61.46±0.01 | 51.28±0.11 | 49.05±36.37 | 43.05±2.19 | 67.16±2.76 | 38.12±5.43 | 81.22±0.57 | / | 54.42±3.86 | 63.87±3.47 | 68.71±0.75 |
| AUC_avg | 57±17.9 | 71.4±0.7 | 75.4±0.2 | 47.6±0.5 | 51.2±32 | 47.4±6.6 | 70.5±3.6 | 51.3±4.3 | 82.1±1.5 | 73.2±7.4 | 68.84±4.74 | 72.8±1.6 | 83±1.8 |

Table 5: Results in F1-score (%) of all 11 compared DNN-based detectors (average of 20 independent trials, "/" indicates that the code provided by the paper cannot run on this dataset).

| Dataset | D_AD | AE | VAE | LUNAR | DROCC | GOAD | N_AD | SOGAAL | REPEN | DAGMM | ICL | flows_ood | nod |
|---|---|---|---|---|---|---|---|---|---|---|---|---|---|
| pima | 34.96±10.24 | 34.4±1.62 | 43.31±0.71 | 34.65±0.21 | 50±25.1 | 31.53±2.01 | 34.33±2.03 | 51.48±0.57 | 51.72±2.33 | 54.34±2.89 | 17.28±3.08 | 55.16±0.65 | 48.99±1.81 |
| breastw | 69.3±20.43 | 50.55±4.67 | 77.58±1.34 | 33.95±0.38 | 48±27.68 | 61.26±2.43 | 46.44±2.48 | 95.21±0.56 | 93.43±1.28 | 48.42±31.71 | 35.58±1.81 | 68.49±1.83 | 94.46±0.9 |
| WBC | 38.5±19.31 | 60±0 | 69.02±1.81 | 10±0 | 53.5±26.51 | 0±0 | 13.5±7.26 | 52.32±2.93 | 72.5±5.36 | 0±0 | 24.24±12.71 | 8.97±0 | 75.5±4.97 |
| wine | 9±22.78 | 10±0 | 9.93±0.17 | 0±0 | 48±28.39 | 14±18.28 | 11±3 | 14.68±0.32 | 90±3.16 | 0±0 | 8.7±13.47 | 15.5±0 | 67±11 |
| HeartDisease | 43.25±14.83 | 34.79±1.53 | 46.59±0.4 | 42.62±0.54 | 41.5±24.35 | 44.08±2.58 | 41.04±2.02 | 38.29±6.85 | 56.62±1.98 | 73.41±2.93 | 22.31±5.41 | 61.19±1.2 | 59.29±3.59 |
| pendigits | 6.86±12.2 | 31.86±1.55 | 33.6±0.34 | 1.92±0 | 47±26.1 | 0.1±0.31 | 6.54±1.74 | 1.96±1.8 | 53.75±4.15 | 2.01±2.23 | 6.6±1.49 | 4.54±0 | 20.06±7.69 |
| Lymphography | 15±18.18 | 83.33±0 | 83.33±0 | 0±0 | 58±25.02 | 5±10.67 | 23.33±14.34 | 71.92±16.02 | 75.83±8.29 | / | 45.71±7.13 | 8.11±0 | 85±10.41 |
| Hepatitis | 19.23±14.49 | 38.46±0 | 38.46±0 | 8.85±7.41 | 41±21.42 | 9.62±9.06 | 8.85±9.51 | 10.77±10.43 | 34.62±12.99 | 24.5±12.41 | 20.95±7.13 | 32.5±0 | 22.69±7.08 |
| Waveform | 5±4.99 | 7.15±0.48 | 6.98±0.03 | 4±0 | 54.5±29.58 | 1.5±1.2 | 13.2±1.96 | 1.64±0.89 | 10.4±2.37 | 5.73±0.09 | 7.37±0.78 | 5.81±0 | 12.5±4.79 |
| wbc | 24.09±18.81 | 47.62±0 | 49.34±1.93 | 5.24±1.43 | 47±25.9 | 0±0 | 23.57±6.11 | 0.48±1.43 | 70.71±1.7 | 1.67±3.97 | 39.32±7.61 | 11.11±0 | 65±4.83 |
| WDBC | 9.5±12.03 | 60±0 | 60±0 | 10±0 | 62±26 | 0±0 | 66.5±9.1 | 5.34±0.03 | 80±0 | 0.27±1.19 | 33.19±3.18 | 5.45±0 | 63.5±4.77 |
| WPBC | 22.02±5.64 | 19.36±0.64 | 14.89±0 | 29.79±0 | 55±32.33 | 24.89±4.37 | 16.91±3.71 | 25.74±5.46 | 21.81±3.29 | 39.82±1.85 | 13.73±3.04 | 38.99±0.49 | 30.32±2 |
| satimage-2 | 27.04±29.61 | 73.66±1.67 | 79.08±0.45 | 3.94±0.56 | 59±27.37 | 27.89±16.37 | 32.56±5.8 | 2.08±0.62 | 91.83±1.81 | 2.2±0.73 | 10.67±2.94 | 2.45±0 | 89.15±1.42 |
| satellite | 35.31±12.49 | 54.9±0.12 | 51.29±0.12 | 32.48±0.09 | 53±26.1 | 36.96±1.43 | 52.73±2.99 | 45.35±6.47 | 57.8±2.98 | 48.24±5.92 | 29.96±8.14 | 50.56±0.45 | 49.72±7.64 |
| KDDCup99 | 10.25±11.99 | 42.28±0 | 42.26±0.03 | 0±0 | 52±32.34 | 35.16±13.21 | 7.24±4.8 | 0.76±0.03 | 1.22±0 | 0.79±0.01 | 6.13±0.9 | / | 37.36±1.5 |
| SpamBase | 41.05±12.14 | 41.94±0.2 | 43.56±0.03 | 39.75±0.15 | 53±30.51 | 36.14±3.11 | 31.5±1.79 | 26.07±3.03 | 45.34±2.55 | 5.73±0.04 | 9.18±1.42 | 46.14±1.85 | 57.33±0.99 |
| optdigits | 2.83±9.71 | 0±0 | 0±0 | 2.17±0.29 | 57±24.1 | 1.1±1.83 | 2.83±1.13 | 3.54±2.45 | 8.6±1.87 | 17.36±0.56 | 6.73±1.43 | / | 5.7±1.67 |
| mnist | 19.42±8.98 | 42.64±0.31 | 39±0 | 9.49±0.22 | 55.5±28.19 | 13.55±4.17 | 47.4±2.42 | 16.66±0.1 | 44.35±1.21 | 17.36±0.56 | 56.4±0.73 | 6.34±0 | 39.79±5.29 |
| musk | 19.54±22.12 | 100±0 | 100±0 | 1.96±0.31 | 53±29.34 | 30.52±22.12 | 88.66±4.08 | 6.14±0 | 89.79±3.34 | 0±0 | 48.02±0 | | 65.15±7.09 |
| Arrhythmia | 55.56±4.44 | 64.08±0 | 64.17±0.19 | 44.66±0.75 | 48.5±22.42 | 40.12±3.04 | 64.25±1.56 | 34.7±2.67 | 64.66±1.33 | 46.87±2.24 | 32.83±0.32 | 68.89±0.63 | 64.49±0.82 |
| speech | 1.72±1.83 | 1.64±0 | 3.28±0 | 3.2±0.36 | 56.5±27.44 | 1.56±1.51 | 3.03±1.19 | 2.47±1.55 | 1.64±0.73 | 3.28±0.04 | 2.33±0.42 | 1.12±0.07 | 1.89±1.4 |
| InternetAds | 40.03±5.27 | 34.24±0 | 34.24±0 | 22.08±0.5 | 50±30.66 | 17.54±1.98 | 30.66±3.85 | 14.17±3.78 | 47.07±0.51 | / | 20.11±4.42 | 33.1±0.27 | 46.37±1.28 |
| F1_avg | 24.98±13.29 | 42.4±0.58 | 44.99±0.34 | 15.48±0.6 | 51.95±27.13 | 19.66±5.44 | 30.27±4.22 | 23.71±3.09 | 52.9±2.87 | 22.04±4.05 | 22.61±3.98 | 27.6±0.39 | 50.06±4.22 |

Table 6: Results in ROC_AUC (%) using different training settings (average of 20 independent trials). (S) means using the original settings of the paper and (U) denotes the unsupervised setting.

| Dataset | DROCC(S) | DROCC(U) | GOAD(S) | GOAD(U) | N_AD(S) | N_AD(U) | LUNAR(S) | LUNAR(U) |
|---|---|---|---|---|---|---|---|---|
| pima | 49.6±12.3 | 48.2±30.2 | 41.5±3.1 | 45.0±2.6 | 60.7±1.3 | 49.9±1.4 | 52.1±0 | 50.5±0.1 |
| breastw | 53.4±34.7 | 46.8±31.9 | 67.9±16.7 | 77.2±3.0 | 96.2±1.0 | 70.4±2.0 | 39.3±0.1 | 49.4±0.1 |
| WBC | 54.2±33.6 | 53.8±31.8 | 24.9±16.2 | 5.7±3.2 | 81.5±4.6 | 85.8±2.3 | 35.3±0.8 | 47.1±0.3 |
| wine | 65.5±32.8 | 47.9±31.6 | 39.0±18.1 | 71.6±20.3 | 95.4±1.9 | 79.3±4.6 | 42.8±1.3 | 30.0±0.6 |
| HeartDisease | 46.7±20.0 | 38.6±26.8 | 43.7±11.1 | 47.9±3.2 | 69.1±4.9 | 46.2±2.4 | 50.1±0.3 | 47.9±0.2 |
| pendigits | 16.7±13 | 44.7±30.4 | 24.8±13.8 | 20.1±12.6 | 98.5±0.8 | 78.6±4.9 | 51.2±0 | 56.4±0.1 |
| Lymphography | 48.8±28.6 | 56.4±31.2 | 98.2±3.6 | 21.4±13.7 | 90.0±4.9 | 83.0±9.3 | 47.8±1.4 | 25.3±1.4 |
| Hepatitis | 55.2±18.4 | 39.7±25.8 | 59.8±10.2 | 39.2±5.6 | 63.3±7.9 | 39.2±11.9 | 55.5±7.1 | 46.5±5.0 |
| Waveform | 49.2±7.4 | 53.8±35.6 | 44.0±2.9 | 44.2±2.2 | 80.1±1.3 | 76.1±2.1 | 48.0±0.2 | 49.5±0.1 |
| wbc | 47.3±30.1 | 45.6±31.0 | 49.5±14.9 | 15.0±3.4 | 92.7±2.0 | 85.7±2.6 | 96.1±0.1 | 42.6±0.3 |
| WDBC | 37.8±34.8 | 62.6±30.1 | 54.8±16.1 | 9.9±3.8 | 97.7±0.6 | 96.6±1.0 | 54.0±0.7 | 47.6±0.2 |
| WPBC | 58.0±8.7 | 54.2±36.6 | 50.3±4.2 | 51.1±2.5 | 49.0±7.1 | 43.9±3.3 | 49.2±0.4 | 47.8±0.2 |
| satimage-2 | 33.4±7.8 | 58.6±30.6 | 98.8±0.6 | 87.8±8.2 | 99.8±0.1 | 97.2±0.7 | 99.9±0 | 55.4±0 |
| satellite | 44.0±1.8 | 50.4±33.6 | 70.8±1.2 | 48.2±2.9 | 81.1±0.4 | 70.2±2.2 | 50.0±0 | 50.9±0 |
| KDDCup99 | 4.4±1.4 | 50.4±36.2 | 91.3±4.6 | 89.4±8.1 | 75.9±12.5 | 76.2±14.4 | 49.6±0 | 50.8±0 |
| SpamBase | 28.3±6.5 | 52.3±35.8 | 40.0±7.4 | 46.2±3.0 | 60.9±3.2 | 39.1±1.9 | 28.8±0.1 | 49.2±0 |
| optdigits | 74.3±13.0 | 56.8±29.1 | 73.5±15.1 | 58.3±13.2 | 82.8±4.6 | 55.0±4.4 | 99.5±0.1 | 48.6±0.2 |
| mnist | 24.1±6.1 | 56.2±33.0 | 56.5±7.0 | 45.0±7.9 | 97.8±0.2 | 88.4±1.3 | 92.4±0.3 | 49.2±0.1 |
| musk | 89.4±5.4 | 54.2±36.1 | 95.1±9.5 | 83.6±16.2 | 99.4±0.1 | 99.8±0.2 | 53.1±0.4 | 47.4±0.2 |
| Arrhythmia | 47.9±9.8 | 48.0±28.0 | 57.6±3.2 | 42.0±3.1 | 69.3±1.8 | 73.6±0.9 | 52.0±0.1 | 48.1±0.4 |
| speech | 58.8±5.5 | 58.2±32.3 | 51.7±4.3 | 51.9±3.7 | 47.9±2.6 | 50.0±1.6 | 49.7±0.2 | 56.8±0.3 |
| InternetAds | 13.6±0.5 | 49.0±36.4 | 52.4±5.2 | 43.0±2.2 | 75.7±1.0 | 67.2±2.8 | 40.8±0.1 | 51.3±0.1 |
| AUC_avg | 45.5±15.1 | 51.2±32.0 | 58.5±8.6 | 47.4±6.6 | 80.2±2.9 | 70.5±3.6 | 56.2±0.6 | 47.7±0.5 |

# E    PERFORMANCE ON DIFFERENT NOISE

To verify the effectiveness of uniformly distributed negative sampling under the NOD framework, we conduct experiments on two commonly used negative sampling methods in the field of outlier detection, SUBSPACE (Goodge et al., 2021) and GAN-BASED (Liu et al., 2020). The SUBSPACE method generates noise by adding Gaussian noise to the subset of feature dimensions of real data. The GAN-BASED method uses GAN to generate noise close to the real data.

To generate uniform noise, we first use uniform probability distribution to generate random values that we named Uniform Random (UR). However, the resulting two negative samples may be very close, giving false signals and disturbing model learning. So we adopt the Fast Poisson Disk (FPD) implementation (Bridson, 2007) to generate negative samples. FPD guarantees that the distance between the two samples is at least user-supplied $r$. But it runs too slowly to generate high-dimensional noise. Thus we only provide results on datasets where $dim$ is below 10 using the FPD method.

In Table 7, we observe that the UR method is more effective than the SUBSPACE and GAN-BASED methods. In addition, the ROC_AUC (AVG_PART) shows no significant performance difference between FPD and UR. Considering the running time in high dimensions, we choose UR as the negative sampling method in NOD.We also incorporated Gaussian noise.NR0.5 means a Gaussian distribution with a mean of 0.5 and a standard deviation of 0.5.NR0.1 means a Gaussian distribution with a mean of 0.5 and a standard deviation of 0.1.The difference between their results is not significant because a Gaussian distribution with a larger std approaches Uniform noise.

# F    ANALYSIS OF DIFFERENT CLASSIFIERS

We evaluate the performance of different optimizers for the binary classification problem. In addition to the SGD, the SVC with RBF kernel (SVC), Decision Tree (DT), and Random Forest (RF) are tested. Fig. 2 shows the results of four different classifiers in the simulated 2-D OD problem. And the ROC_AUC performance of different classifiers is shown in Table 8.

Although most machine learning algorithms are designed with the so-call "smoothness prior", i.e., the function learn should not vary very much within a small region (Goodfellow et al., 2016), their actual performance in this binary classification task is quite different. As shown in Fig. 2, SVC_RBF, DT and RF try to separate different regions between positive samples and generated noise points with rigid boundaries. However, the inliers may overlap or be close to the random noise points. Thus,

Table 7: ROC_AUC (%) performance under different noise.("/" indicates that the method did not obtain results within 2 hours)

| Dataset | SUBSPACE | GAN | FPD | NR0.5 | NR0.1 | UR |
|---|---|---|---|---|---|---|
| pima | 54.5 | 26.9 | 61.9 | 62.9 | 62.5 | 62.9 |
| breastw | 47.1 | 5.8 | 99.4 | 99.4 | 98.7 | 99.3 |
| WBC | 42.4 | 18.8 | 99.2 | 99.2 | 98.7 | 99.2 |
| wine | 44.3 | 10.8 | / | 93.8 | 94.4 | 97.2 |
| HeartDisease | 44.7 | 17.7 | / | 64.5 | 70.1 | 67.1 |
| pendigits | 56.9 | 79.2 | / | 91.8 | 83.3 | 91.6 |
| Lymphography | 56.2 | 48.4 | / | 99.8 | 99.3 | 99.8 |
| Hepatitis | 47.2 | 71.8 | / | 64.4 | 73.3 | 69.7 |
| Waveform | 51.6 | 57.3 | / | 80.8 | 79.6 | 80.7 |
| wbc | 50.1 | 1.5 | / | 95.2 | 95.6 | 95.9 |
| WDBC | 55.5 | 2.1 | / | 97.3 | 97.2 | 97.4 |
| WPBC | 50.4 | 44.4 | / | 57.9 | 57.0 | 57.8 |
| satimage-2 | 48.3 | 73.1 | / | 99.0 | 99.4 | 99.5 |
| satellite | 48.7 | 72.3 | / | 78.9 | 68.6 | 74.4 |
| KDDCup99 | 46.7 | 96.2 | / | 99.0 | 98.8 | 98.9 |
| SpamBase | 52.5 | 38 | / | 67.9 | 67.6 | 68.4 |
| optdigits | 50.4 | 51.8 | / | 76.0 | 78.5 | 76.2 |
| mnist | 45.8 | 82.2 | / | 86.7 | 83.6 | 86.3 |
| musk | 53.6 | 99.2 | / | 99.3 | 90.8 | 98.2 |
| Arrhythmia | 51.4 | 68.9 | / | 74.0 | 73.7 | 74 |
| speech | 49.4 | 49.2 | / | 60.8 | 58.7 | 62 |
| InternetAds | 51.8 | 40.6 | / | 68.7 | 68.6 | 68.7 |
| lympho | 52.6 | 81 | / | 97.7 | 96.2 | 97.3 |
| arrhythmia | 49.7 | 78.3 | / | 77.9 | 77.7 | 77.8 |
| vowels | 44 | 20.1 | / | 60.7 | 70.3 | 63.5 |
| letter | 50.7 | 45.8 | / | 59.7 | 58.8 | 58.9 |
| cardio | 58.1 | 70.9 | / | 80.4 | 74.3 | 77.8 |
| mammography | 43 | 36 | 84.9 | 84.8 | 74.6 | 81.7 |
| shuttle | 35.9 | 38.8 | / | 91.2 | 94.6 | 95.7 |
| Stamps | 48.8 | 67.7 | 91.3 | 90.6 | 84.6 | 88.6 |
| Pima | 46.5 | 29 | 62.8 | 63.7 | 60.9 | 63.4 |
| AUC_avg | 49.3 | 49.2 | / | 82.6 | 81.7 | 83.0 |
| AUC(PART) | 47 | 30.7 | 83.3 | 83.4 | 81.6 | 82.5 |

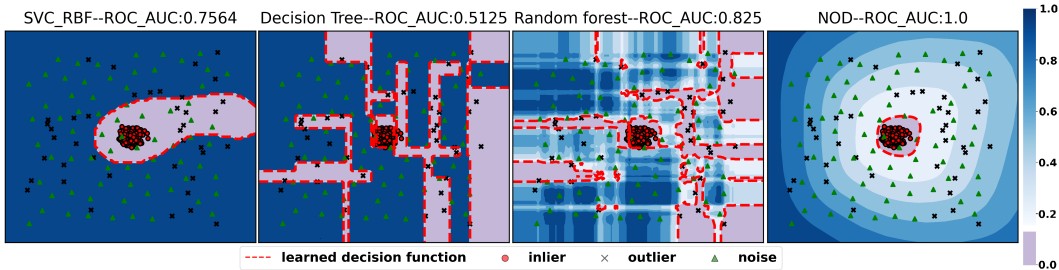

Figure 2: Comparison of different classifiers.

Table 8: ROC_AUC (%) performance under different classifiers.

| Dataset | Linear_SVC | DT | RF | LGB | SGD | ADAM |
|---|---|---|---|---|---|---|
| pima | 53.7 | 50.2 | 50.3 | 55.5 | 62.9 | 58.9 |
| breastw | 66.0 | 50.3 | 50.6 | 48.1 | 99.3 | 98.6 |
| WBC | 83.2 | 55.0 | 55.0 | 91.2 | 99.2 | 99.2 |
| wine | 81.0 | 49.8 | 49.6 | 67.6 | 97.2 | 93.7 |
| HeartDisease | 53.5 | 50.0 | 50.0 | 50.0 | 67.1 | 63.0 |
| pendigits | 63.5 | 50.3 | 50.0 | 59.8 | 91.6 | 87.0 |
| Lymphography | 98.1 | 50.0 | 50.0 | 98.9 | 99.7 | 99.9 |
| Hepatitis | 49.1 | 50.0 | 50.0 | 40.0 | 69.7 | 53.9 |
| Waveform | 75.1 | 50.2 | 50.3 | 59.7 | 80.7 | 82.1 |
| wbc | 64.2 | 52.2 | 52.2 | 83.8 | 95.9 | 84.6 |
| WDBC | 74.3 | 56.7 | 60.0 | 82.3 | 97.4 | 93.5 |
| WPBC | 48.6 | 50.2 | 49.5 | 52.2 | 57.8 | 55.3 |
| satimage-2 | 96.0 | 50.9 | 50.6 | 96.6 | 99.5 | 99.7 |
| satellite | 59.2 | 50.0 | 50.0 | 52.2 | 74.4 | 48.4 |
| KDDCup99 | 50.0 | 50.0 | 50.0 | 50.0 | 98.9 | 98.8 |
| SpamBase | 50.0 | 50.0 | 50.0 | 50.0 | 68.4 | 65.8 |
| optdigits | 50.0 | 50.0 | 50.0 | 50.0 | 76.2 | 70.5 |
| mnist | 50.0 | 50.0 | 50.0 | 50.0 | 86.3 | 88.1 |
| musk | 50.0 | 50.0 | 50.0 | 54.4 | 98.2 | 91.5 |
| Arrhythmia | 50.0 | 49.8 | 50.0 | 38.2 | 74.0 | 75.8 |
| speech | 49.9 | 49.9 | 50.0 | 49.1 | 62.0 | 60.8 |
| InternetAds | 50.0 | 49.9 | 50.0 | 51.5 | 68.7 | 57.8 |
| AUC_avg | 62.1 | 50.7 | 50.8 | 60.5 | 83.0 | 78.5 |

these classifiers cannot produce a smooth distribution estimation with their hard separation methods. SGD, in contrast, can generate smooth boundaries with different levels of abnormality. As seen in NOD, the center of the cluster has a very low anomaly score, and we have high anomaly scores when there are fewer inliers or outliers. In practice, there is often no clear boundary between outliers and outliers. Therefore, our solution can provide more detailed information about the degree of sample abnormality than solutions with only 0,1 labels.

## G  EXAMPLE OF MULTIPLE CLUSTERING CENTERS ON TWO-DIMENSIONAL DATA

As shown in Fig.3, we constructed some two-dimensional composite datasets with multiple clustering centers and visualized the distribution of NOD anomaly scores on each dataset. From Fig.3, it can be seen that under the premise of complying with the basic assumption of NOD, NOD is effective on datasets with multiple clustering centers.

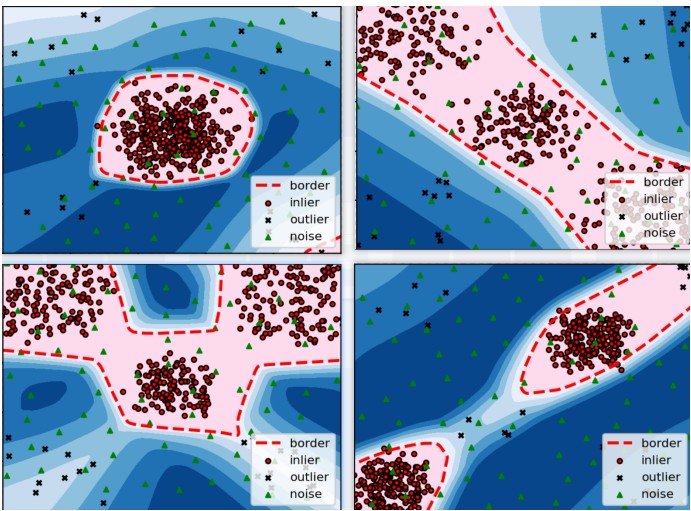

Figure 3: Example of multiple clustering centers on two-dimensional datasets.

## H PERFORMANCE OF THE DIFFERENT EMBEDDING METHODS IN IMAGE DATASETS

Table 9 shows the anomaly detection performance of NOD on anomaly detection datasets constructed using different image embedding methods. From the experimental results in the table, it can be seen that different embedding methods can seriously affect the performance of NOD. Different pre-training models have different capabilities to capture intricate presentations or patterns. Resnet152 is much stronger than resnet18. Therefore, Resnet152 embeds more information than resnet18. Thus, the embeddings of outliers from resnet18, due to its lack of interacted patterns, are much more clustered than the ones from resnet152. This might explain the huge performance difference between resnet18 and resnet152 while the small performance difference between resnet50 and resnet152. Therefore, extending NOD to end-to-end anomaly detection solutions is a direction that needs to be explored in the future.

Table 9: ROC_AUC(%) performance of the different embedding methods on Image datasets.

| Dataset | resnet18 | resnet50 | resnet152 |
|---|---|---|---|
| airplane | 68.22±0.79 | 91.34±0.18 | 95.31±0.07 |
| automobile | 42.75±0.91 | 95.94±0.05 | 96.62±0.08 |
| bird | 57.41±0.57 | 85.56±0.02 | 87.95±0.21 |
| cat | 46.83±1.00 | 88.87±0.02 | 90.01±0.13 |
| deer | 74.21±0.35 | 92.91±0.03 | 95.62±0.09 |
| dog | 41.20±1.02 | 88.95±0.32 | 92.28±0.26 |
| frog | 63.97±0.92 | 95.42±0.10 | 96.50±0.06 |
| horse | 53.54±0.59 | 91.14±0.25 | 95.19±0.11 |
| ship | 65.02±0.70 | 96.20±0.07 | 97.00±0.10 |
| truck | 57.83±0.91 | 96.89±0.10 | 97.66±0.01 |
| AUC_avg | 57.10±0.78 | 92.23±0.11 | 94.41±0.11 |

## I PERFORMANCE COMPARISON BETWEEN NOD AND SOME DENSITY-BASED METHODS

NOD is suitable for density-based scenarios. Here, some classical density-based methods are involved in comparisons including LOF(Breunig et al., 2000), CBLOF(He et al., 2003), COF(Tang et al., 2002) and LOCI(Papadimitriou et al., 2003). Compared to these methods, NOD still has a significant performance lead.

Table 10: ROC_AUC(%) performance comparison between NOD and some density-based methods. ("/" indicates that the method did not obtain results within 2 hours, OOM denotes the out-of-memory error with 512G memory)

| Dataset | LOF | CBLOF | COF | LOCI | NOD |
|---|---|---|---|---|---|
| pima | 53.84 | 60.52 | 51.86 | 44.45 | 62.9 |
| breastw | 38.32 | 96.27 | 33.22 | 17.03 | 99.29 |
| WBC | 83 | 98.73 | 73.94 | 86.2 | 99.16 |
| wine | 99.75 | 99.92 | 97.9 | 65.46 | 97.21 |
| HeartDisease | 50.05 | 57.92 | 52.7 | 35.35 | 67.08 |
| pendigits | 47.94 | 92.2 | 52.37 | / | 91.59 |
| Lymphography | 97.65 | 99.88 | 99.41 | 83.92 | 99.74 |
| Hepatitis | 62.57 | 63.61 | 51.09 | 39.27 | 69.69 |
| Waveform | 73.41 | 74.97 | 70.03 | / | 80.74 |
| wbc | 92.97 | 94 | 87.13 | / | 95.9 |
| WDBC | 98.15 | 98.18 | 99.1 | 78.99 | 97.39 |
| WPBC | 51.85 | 46.78 | 47.43 | 43.61 | 57.76 |
| satimage-2 | 53.25 | 99.86 | 55.83 | / | 99.51 |
| satellite | 53.95 | 73.2 | 53.55 | / | 74.43 |
| KDDCup99 | 62.54 | OOM | 60.86 | / | 98.94 |
| SpamBase | 45.13 | 55.08 | 43.49 | / | 68.4 |
| optdigits | 58.79 | 88.28 | 57.29 | / | 76.15 |
| mnist | 64.49 | 80.43 | 62 | / | 86.3 |
| musk | 41.24 | 100 | 40.7 | / | 98.18 |
| Arrhythmia | 72.59 | 73.45 | 71.91 | 64.95 | 73.98 |
| speech | 50.87 | 47.28 | 52.98 | / | 62.02 |
| InternetAds | 65.54 | 71.42 | 67.86 | / | 68.71 |
| AUC_avg | 64.45 | 79.62 | 62.85 | 55.92 | 82.96 |

## J  PARAMETER SENSITIVITY ANALYSIS

This section examines the effects of various settings in NOD, including the ratios of negative samples, hidden layer dimensions, number of layers, and the usage of early stopping. Fig. 4(a) shows relative ROC_AUC change rates (Y-axis) with different ratios of negative samples (e.g., $0.1 * |X|$). X-axis values denote the dataset index ordered as Table 1. $|X|$ is normalized to 1 for the 22 datasets. Dots above 1 mean improved performance, while those below 1 indicate underperformance. We observe that the performance generally deteriorates when there are too many/small negative samples (e.g., the brown dots). Fig. 4(b) shows the effect of varying the hidden layer dimension. NOD is insensitive to changes in the hidden layer dimension. Fig. 4(c) shows the impacts of the number of layers with average ROC_AUC and standard deviations across 20 runs. The results indicate that a model with two layers outperforms a single-layer model. However, as the number of layers increases, the model's fitting ability increases while the risk of overfitting also rises. Fig. 4(d) shows the comparison with(out) the proposed early stop. For most datasets, the early stop can effectively reduce the impact of overfitting and achieve better performance.

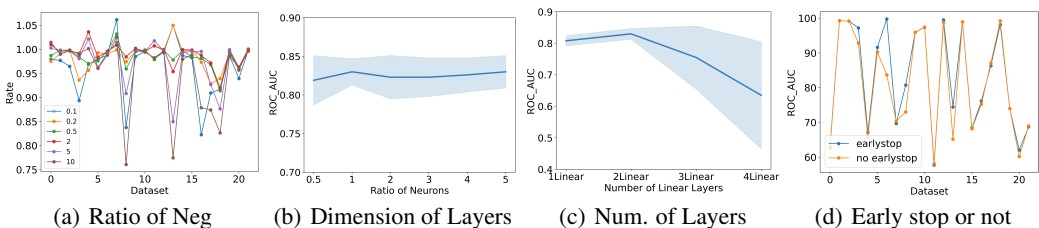

| (a) Ratio of Neg | (b) Dimension of Layers | (c) Num. of Layers | (d) Early stop or not |

Figure 4: Performance under different settings. Shaded areas indicate standard deviations. X axis of (a) and (d) is the number of datasets with the same order as Table 2
.

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
