# OpenReview forum: "Noise-guided Unsupervised Outlier Detection"
_ICLR.cc/2024/Conference — Submitted to ICLR 2024_

### Official Review · Reviewer_35WP · 2023-10-25

**Soundness:** 1 poor
**Presentation:** 1 poor
**Contribution:** 2 fair
**Rating:** 3
**Confidence:** 4

**Summary:**

This paper proposes an unsupervised learning-based approach for outlier detection. Specifically, they propose to train a binary anomaly classifier discriminating the given data and the uniform noise. They provide a theoretical analysis supporting the existence of an optimal classifier. To validate their approach, they conducted experiments on 22 different datasets and compared the performance to state-of-the-art baseline methods.

**Strengths:**

- The introduction effectively conveys a compelling motivation.
- The suggested approach is straightforward to put into practice.
- The proposed method achieves state-of-the-art performance on diverse real-world datasets.
- The authors conduct a variety of analyses on their proposed method, making them valuable.

**Weaknesses:**

- The theory is challenging to comprehend as the explanations are insufficient and lack consistency.
   - For instance, in Lemma 1, they introduce the density function $\rho$, but it's not clear what the underlying distribution of this density function is and its conceptual interpretation.
   - The proof provided in the appendix is complex to follow and is not self-contained. (please see the questions below)
- Figure 1 is unclear and lacks adequate explanation.
- There are numerous inconsistent expressions and typos:
  - p.4 Proposition 1 => Assumption 2
  - p.4 socre => score
  - p.4,5 Equ. => Eq.
  - p.5 donot => do not
  - In the proof of lemma 1, x,y => $x,y$
  - In the proof of lemma 1, they use $y$ to denote a data point, but in p.3, $y$ stands for the label.
- The uniform sampling approach may have limitations in high-dimensional data.

**Questions:**

- In lemma 1, what is the underlying distribution of $\rho$?
- In the proof of lemma 1 in the appendix, what is the definition of the scaling factor?
- In the proof of lemma 1 in the appendix, what is the source of the term $4\sqrt{dim}$?
- In the proof of lemma 1 in the appendix, what is the meaning of “dataset space”?
- In the proof of lemma 1 in the appendix, what is the meaning of “tag”?
- In figure 1, why assigning the same score to A,B, and C is a disadvantage and limitation of the kNN approach? From the figure, it seems that the kNN approach is able to classify all three points as outliers.
- Can the approach be applied to high-dimensional data, such as ImageNet?

---

### Official Review · Reviewer_8pA6 · 2023-10-28

**Soundness:** 2 fair
**Presentation:** 2 fair
**Contribution:** 2 fair
**Rating:** 3
**Confidence:** 4

**Summary:**

The paper proposes NOD, a method for unsupervised outlier detection. NOD trains a binary classifier to distinguish between sample data and generated noise. The score produced by this classifier is considered as the anomaly score of data samples. NOD is backed by some theoretical results.

**Strengths:**

NOD, being a binary classifier, is very simple and easy to deploy. This simplicity is based on the following reasonings (verbatims of the original text in the paper):
- outliers should be generally sparser distributed than inliers,
- noises with uniform prior probability density have the maximum entropy, and thus essentially have little chance of resembling structured inputs, hence should have high probability of being outlier than most samples in the original (unlabelled) data, and
- random noise can act as a trustworthy and stable reference for a sample's anomaly degree in the whole data space and demands few assumptions towards the inliers' distribution

Based on the above observations, the authors propose to generate random noises and train the classifier to differentiate between original data samples and noises. My understanding here is this model will assign outliers in the original data higher anomaly score than inliers.

NOD is corroborated by a couple of theoretical results.

Experimental results look comprehensive and solid.

**Weaknesses:**

From Equations (1) & (2) and Figure 2, one can see that both inliers and outliers in the original data sets are used as positive samples in the training set. But then would this be an issue? Put differently, I don't follow how the binary classifier can distinguish original outliers from inliers given that both belong to the same class in training data.

Propositions, lemmas and theorems are hard to follow. I suggest to have a table of notations used.

The proofs are also not that convincing. For instance, Lemma 1 sounds as if it's applicable to any type of noise. The proof in the Appendix needs specifically generated uniform noise. To fix this, please restate this Lemma (and others) to make clear what kind of inputs are required, e.g. noises generated in a specific way.

**Questions:**

How can the binary classifier differentiate between inliers and outliers?

The propositions, lemmas and theorems in the paper need to be revised to include all necessary inputs.

In the experiments, NOD is run 20 times. Is this to account for fluctuation between different training rounds?

---

### Official Review · Reviewer_DJGq · 2023-11-01

**Soundness:** 3 good
**Presentation:** 3 good
**Contribution:** 3 good
**Rating:** 5
**Confidence:** 4

**Summary:**

The paper proposes NOD, an anomaly detection method for unsupervised settings. NOD focuses on the idea that instead of separating outliers from inliers, we can actually compare data against uniform random noise to spot changes from normal behavior. Experimental results across several datasets and baselines demonstrate the potential of this solution.

**Strengths:**

- Timely and challenging problem, especially under unsupervised settings
- Idea is nicely motivated and appears to work well
- Experimental results support the overall claims of this work

**Weaknesses:**

- Novelty is somewhat low
- Experimental settings not clear for baselines
- Experimental results are not outperforming clearly baselines

**Questions:**

- Novelty is somewhat low

The idea is nicely motivated at first, but then the novelty becomes unclear. Several earlier solutions (REPEN) seems to rely on similar ideas so this appear to not be novel. The negative examples generation, smoothness prior etc. all components already exist. The work needs to highlight new ideas vs. existing ones and preliminaries can help understand how these solutions build on top of prior work

- Experimental settings not clear for baselines

Settings for baselines are not clear. The work indicates that they use a particular package and parameters are optimized for these datasets, but these are not clear settings for reproducibility. Clear settings, parameter choices, estimation techniques, etc. need to be provided for the baselines

- Experimental results are not outperforming clearly baselines

Experimental results do not show significant improvement over prior solutions relying on a similar concept (comparison against noise) - REPEN is the second best method and results appear to not be statistically significantly different.

---

### Meta-Review · Area_Chair_3Ppt · 2023-11-29

**Metareview:**

I have read all the materials of this paper including the manuscript, appendix, and comments (no response was provided by the authors). Based on collected information from all reviewers and my personal judgment, I can make the recommendation on this paper, *rejection*.

**Research Question**

This paper considers the well-defined outlier detection problem.

**Challenge Analysis**

The authors claim that finding a clear boundary between inliers and outliers is challenging.

**Philosophy**

The authors aim to use some synthetic outliers to facilitate the boundary learning. Specifically, the authors believe a random point has a high probability to be an outlier.

**Related Work**

One reviewer pointed out there exist several studies that employing the generated samples (self-supervised fashion) for outlier detection. It is necessary to illustrate and demonstrate the benefits of the proposed method over them.

**Techniques**

The authors proposed the Noise-guided unsupervised Outlier Detection (NOD), the core idea of which is to train a classifier to separate the original dataset (contains inliers and outliers as the positive class) with a bunch of random generated points (as the negative class).

In essences, the formulated learning task is a binary classification with noisy labels. It is suggested to employ some noisy label techniques, for example, early stop (I noticed that authors mentioned the early stop in their experimental setup) or modified loss functions/regularizer.

**Random Samples and High Dimensions**

It is difficult to generate random samples uniformly covering the whole space. If so, the number of random samples is quite huge. Currently, the number of random samples is equal to the number of points in the original dataset. This raises a question that whether these random samples are enough. The discussion or exploration on negative samples are insufficient. High dimensionality is another issue to affect the number of random samples.

**Definition of Outliers**

Some outliers are clustered together, such as clustered outliers, which are not well captured by the proposed method. This is fine. It is suggested to give a formal definition of outliers that this paper aims to address. It is quite important to know when the proposed method works and when not.

**Experiments**

1. The experimental results seem promising.

2. More explorations on random samples are needed.

This is the second time to review this paper as an AC, where I can see some modifications in the current version. In general, I like simple, neat, and elegant ideas, which is easy to use in practice, which is the merit of this paper. However, more improvements (studies in the same category, noisy label learning, and random sample exploration) are needed to make this paper more stronger.

**Justification For Why Not Higher Score:**

1. All reviewers and I have a consensus reject recommendation on this paper.
2. The authors did not provide the response.

**Justification For Why Not Lower Score:**

N/A

---

### Decision · Program_Chairs · 2024-01-16

Reject